# Interplay of Ferritin Accumulation and Ferroportin Loss in Ageing Brain: Implication for Protein Aggregation in Down Syndrome Dementia, Alzheimer’s, and Parkinson’s Diseases

**DOI:** 10.3390/ijms23031060

**Published:** 2022-01-19

**Authors:** Animesh Alexander Raha, Anwesha Biswas, James Henderson, Subhojit Chakraborty, Anthony Holland, Robert P. Friedland, Elizabeta Mukaetova-Ladinska, Shahid Zaman, Ruma Raha-Chowdhury

**Affiliations:** 1John van Geest Centre for Brain Repair, Department of Clinical Neuroscience, University of Cambridge, Cambridge CB2 2PY, UK; alex.raha@icloud.com (A.A.R.); jh2014@cam.ac.uk (J.H.); subhojit.chakraborty@ucl.ac.uk (S.C.); 2Department of Medicine, University of Cambridge, Cambridge CB2 0QQ, UK; 3Department of Biochemistry, The M. S. University of Baroda, Vadodara 39002, India; anwesha611@gmail.com; 4NIHR Biomedical Research Centre, Moorfields Eye Hospital and UCL Institute of Ophthalmology, London EC1V 9EL, UK; 5Cambridge Intellectual & Developmental Disabilities Research Group, Department of Psychiatry, University of Cambridge, Cambridge CB2 8AH, UK; ajh1008@medschl.cam.ac.uk (A.H.); shz10@medschl.cam.ac.uk (S.Z.); 6Department of Neurology, School of Medicine University of Louisville, Louisville, KY 40292, USA; robert.friedland@louisville.edu; 7Department of Neuroscience, Psychology and Behaviour, University of Leicester, Leicester LE1 7RH, UK; eml12@leicester.ac.uk

**Keywords:** basal ganglia, locus coeruleus, substantia nigra, striosomes/matrix, neurodegeneration, ferritin, hepcidin, Alzheimer’s disease (AD), Parkinson’s disease (PD), Down syndrome (DS)

## Abstract

Iron accumulates in the ageing brain and in brains with neurodegenerative diseases such as Alzheimer’s disease (AD), Parkinson’s disease (PD), Huntington’s disease (HD), and Down syndrome (DS) dementia. However, the mechanisms of iron deposition and regional selectivity in the brain are ill-understood. The identification of several proteins that are involved in iron homeostasis, transport, and regulation suggests avenues to explore their function in neurodegenerative diseases. To uncover the molecular mechanisms underlying this association, we investigated the distribution and expression of these key iron proteins in brain tissues of patients with AD, DS, PD, and compared them with age-matched controls. Ferritin is an iron storage protein that is deposited in senile plaques in the AD and DS brain, as well as in neuromelanin-containing neurons in the Lewy bodies in PD brain. The transporter of ferrous iron, Divalent metal protein 1 (DMT1), was observed solely in the capillary endothelium and in astrocytes close to the ventricles with unchanged expression in PD. The principal iron transporter, ferroportin, is strikingly reduced in the AD brain compared to age-matched controls. Extensive blood vessel damage in the basal ganglia and deposition of punctate ferritin heavy chain (FTH) and hepcidin were found in the caudate and putamen within striosomes/matrix in both PD and DS brains. We suggest that downregulation of ferroportin could be a key reason for iron mismanagement through disruption of cellular entry and exit pathways of the endothelium. Membrane damage and subsequent impairment of ferroportin and hepcidin causes oxidative stress that contributes to neurodegeneration seen in DS, AD, and in PD subjects. We further propose that a lack of ferritin contributes to neurodegeneration as a consequence of failure to export toxic metals from the cortex in AD/DS and from the substantia nigra and caudate/putamen in PD brain.

## 1. Introduction

For all animals including humans, regulatory mechanisms have evolved to ensure that iron homeostasis is maintained both at the whole-body and at the cellular level [1,2,3]. Iron pathways are critical for normal brain function and needed for the synthesis and metabolism of many neurotransmitters including dopamine (DA), norepinephrine, and serotonin [4]. The brain has high energy demands and iron is required by brain mitochondria to generate adenosine triphosphate (ATP) by electron transport. Dietary iron deficiency is associated with cognitive and motor impairment [4,5,6]. There are iron rich areas in the brain, such as the substantia nigra, red nucleus, basal ganglia, and deep cerebellar nucleus [7,8]. Uptake, distribution, and sequestration of iron are regulated at the cellular level by transferrin (Tf), its receptor (TfR), and ferritin [9,10]. Ferritin is an iron storage protein, and stores iron in the glial cells (including astrocytes and oligodendrocytes) in the brain. There are three different isoforms of ferritin, light chain (FTL), heavy chain (FTH), and mitochondrial ferritin (MtF) [11,12,13]. All three types of ferritin isoform have distinct role in iron metabolism in the brain [13,14]. The transferrin receptor plays a key role for cellular iron uptake and must be transported across the endosomal membrane to be released into the cytosol [10]. This transport is mediated by the divalent metal transporter 1 (DMT1), also known as Nramp2, a conserved membrane protein [15,16]. Ferroportin is a transmembrane protein that exports iron from cells to plasma [17,18,19]. It is localised in most cell types within the brain including neuronal perikarya, axons, dendrites, and synaptic vesicles.

Hepcidin is a key regulator of iron absorption which is involved in whole body iron homeostasis by controlling iron flux into the plasma from the duodenum as well as iron recycling macrophages through binding to its receptor, the ferroportin [20,21]. This process also depends on the regulation of other metals such as copper and zinc and their transporters (cu/zinc dismutase, hephaestin, and ceruloplasmin) [22]. It was reported that amyloid precursor protein (APP) may bind to ferroportin to facilitate neuronal iron export and that disturbances in this process may be implicated in AD [23].

AD is the most common age-related neurodegenerative disease, characterised by cerebrovascular and neuronal dysfunction leading to progressive decline in cognitive functions and the development of dementia [24,25,26,27]. Pathological hallmarks of AD include senile plaques (SPs), primarily composed of amyloid beta peptide (Aβ42), and neurofibrillary tangles (NFTs) consisting of hyper-phosphorylated microtubule-associated tau protein [28]. Another neurological disease is Down syndrome (DS) dementia, which also develops the clinical and neuropathological features of AD. The likely cause of AD in DS is due to the presence of an extra copy of chromosome 21 (Hsa21) where the APP gene is located [29,30,31,32]. All adults with DS over the age of 40 years display neuropathological hallmarks of AD, including SPs and NFTs [33,34]. High Aβ42 accumulation is toxic and could enhance the formation of reactive oxygen species (ROS) and oxidative stress in neurons leading to neuroinflammation and premature cell death [35,36,37]. Aβ plaque deposition is an early event seen in DS brain. Aβ and the iron storage protein ferritin have been shown to co-localise in the vascular amyloid deposits of plaque in post-mortem AD/DS brains [26,38,39,40]. In contrast, Parkinson’s disease is neuro-pathologically characterised by the progressive degeneration and subsequent loss of dopaminergic (DA) neurons of the substantia nigra pars compacta (SNpc) in the basal ganglia [41,42]. The DA neurons of SNpc extend their fibers to the caudate-putamen, forming the nigrostriatal pathway. This pathway is essential for voluntary movements in normal subjects [43,44]. The population of pigmented neurons composed of neuromelanin in substantia nigra (SN) is the main target of the neurodegenerative process in PD [45,46]. The neurons die in a slow and progressive manner, causing a decrease in dopamine content of the striatum, leading to loss of the ability to control voluntary movements [47]. Another hallmark of PD is the formation of proteinaceous cytoplasmic inclusions, known as Lewy bodies, that accumulate iron and α-synuclein [48,49]. Iron and its major molecular forms, such as ferritin and neuromelanin (NM), have been found in SN and locus coeruleus (LC) of normal subjects at various ages [50,51].

To uncover the molecular mechanisms underlying these associations, we investigated and compared the distribution and expression of key iron proteins ferritin, ferroportin, DMT1, and hepcidin in human brain tissues from patients with AD, DS, PD, and age-matched controls. We also analysed embryonic and adult mouse/rat tissues when human brain tissues were unavailable for appropriate analysis. We previously reported extensive blood vessel damage and reduction in hepcidin and ferroportin levels in AD brain [26]. In this manuscript, we explore the expression of iron related proteins in the basal ganglia, SNpc, and LC. Our results suggest that a downregulation of ferroportin leads to iron accumulation in the SN and caudate/putamen via disruption of the cellular entry and exit pathways. We further propose that iron-induced oxidative stress contributes to neurodegeneration as a result of failure to export toxic metals from the blood vessels in the brain. Additionally, accumulation of iron in the endosomes and striosomes leads to protein aggregation found in the SP, NFT, and Lewy body. Failure to clear excess iron by microglia increases Lewy body formation in PD. Furthermore, astrocytes fail to clear iron molecules that bind with amyloids and leads to plaque formation.

## 2. Results

### 2.1. Non-Haem Iron Concentration Was Higher in the PD Brains

Brain tissue samples from substantia nigra pars compacta (SNpc) and locus coeruleus (LC) from AD, PD, DS, and age-matched controls (*n* = 4 in each group) were collected during autopsy as detailed in the methods section and listed in Table 1. Brain sections from each subject were homogenised in a plastic tube (using fast prep poly-lysin beads). To measure non-haem iron content, we followed the modified method of Torrance and Bothwell, using ferrozine as the chromogen [52]. All results are reported as μg iron/g wet weight as shown in Table 2. The median non-haem iron concentration in the SNpc of old control (OC) subjects was 254 μg/g wet weight, significantly higher than the median of young controls (YC) of 189 μg/g wet weight. The highest non-haem iron concentration in the SNpc was found in PD subjects (337 μg/g wet weight), followed by AD (261 μg/g wet weight), and DS subjects (209 μg/g wet weight), the latter being slightly higher than young controls (Figure 1A, Table 2). The concentration of iron in LC was significantly lower in all four groups (~150 μg/mg wet tissues, Figure 1B, Table 2) compared to that in SNpc and remained constant throughout (Figure 1A–C). The concentration of iron in SNpc increases with age unlike that seen in the LC (*R*^2^ = 0.76, *p* < 0.0001) (Figure 1D, Table 2).

### 2.2. Lewy Bodies Were Visible in the Neuromelanin Cells of PD Brain

Neuromelanin is an insoluble pigment found in neurons of specific brain regions, especially in the SNpc and LC. Brain sections (from SNpc and LC) from AD, PD, DS, and age-matched controls *(n* = 4) were stained with Pearls’ stain (and counter stained with eosin) that could recognise ferric iron deposition in the brain tissues and then analysed using a light microscope. In the SNpc of control subjects, a significant amount of black granular iron molecules was visible in the neuromelanin-containing neurons close to the blood vessels (Figure 1E–G) and with lower expression visible in the damaged neuromelanin cells of PD subjects (Figure 1H). Similarly, intact neuromelanin cells with processes were visible in the AD brains without any cell damage (Figure 1I,J). Neuromelanin content in the LC neurons were much lower than in the SNpc in the control brains and even lower in the DS subjects (Figure 1K,L). To evaluate iron content in other parts of the brain, a cortical section from AD cortex was stained with Pearls’ stain and ferric iron accumulation was seen in the SPs (Figure 1M). We further analysed brain sections from PD subjects with human α-synuclein antibodies. We found α-synuclein protein located in the neuromelanin cells in the DS brain (Figure 1N) without any Lewy body but high numbers of lipofuscin molecules around the neuromelanin cells. In the PD SNpc, α-synuclein protein was located in the neuromelanin neurons and Lewy bodies were present inside the neuromelanin cells (Figure 1O,P). A considerable amount of lipofuscin granules was observed to be scattered close to the Lewy bodies (Figure 1O,P). These neuromelanin granules might be a contributor of oxidative stress and could be toxic to brain cells.

### 2.3. Ferroportin mRNA Expression Is Downregulated in PD

We examined mRNA expression of FTL, FPN, and DMT1 in human AD, DS, PD, and age-matched control brain tissues by in situ hybridisation (n = 6 in each group). Ferritin mRNA was present in the hippocampus, dentate gyrus granule cells and cerebellum in the control brains (Figure 2A,B), whereas less mRNA expressed in the hippocampus of AD subjects (Figure 2E). In controls, particularly in the SNpc, large number of neuromelanin cells showed a strong hybridisation signal (silver grains) (Figure 2C,D). The highest amount of FTL stain was seen in the SNpc of normal aged brain (Figure 2C,D) and then in the PD and AD brain neurons (Figure 2F,H), whereas the DS brain sections showed very low FTL mRNA expression, most of the neuromelanin cells were empty (Figure 2G).

Ferroportin was visible in the neuromelanin cell membrane (Figure 2I–L). Interestingly, the ferroportin probe was bound to the axon initial segment (AIS) of the neuromelanin containing neurons (Figure 2J–L). Ferroportin mRNA expression was very limited in the PD and DS brains and was only detected in the AIS of the neuromelanin cells (Figure 2J–L). Ferroportin mRNA expression was less in the LC when compared to the SNpc (Figure 2M–P). There was minimal difference in DMT1 expression in the SNpc or LC of PD or in the control brain (Figure 2Q–T). DMT1 was also expressed in the AIS of the neuromelanin cells (Figure 2R,T) suggesting that both genes were transcribed in neuromelanin cells where they might function as solute carriers. To validate stringency of mRNA expression, two PD brain samples were hybridised with α-synuclein probes, which was highly expressed in the SNpc of PD brain (Figure 2U,V). A β-actin probe was used as a positive control showing strong expression in the cerebellum (Figure 2W) and another liver section was probed with FTL to prove hybridisation stringency and as highest amount of FTL mRNA is synthesised in the liver (Figure 2X). FPN and DMT1 mRNA is expressed mainly in the gut endothelium, and when mouse gut sections were probed with ferroportin, DMT1, hephaestin, and ceruloplasmin, results showed mRNA expression in the gut mucosa (Figure 2Y,Z(i–iii)).

### 2.4. Ferroportin Protein Expression Was Lower in PD Brain Compared to Control and DS Brain

To establish whether the changes observed in transcription of iron regulating proteins were also seen at the level of protein translation, Western blotting (WB) was performed on PD, DS, and age-matched control samples (*n* = 6) from the basal ganglia. We have previously described ferroportin protein expression in the AD brain [26]. Lysate was prepared from basal ganglia tissues (both caudate and putamen), separated in appropriate pore size in poly-acrylamide gel, and then transferred in PVDF membrane (as described in the methods) and probed with anti-ferritin light chain, molecular weight (MW), (FTL, 19 kDa), anti-ferritin heavy chain (FTH, 21 kDa), anti-MtF (22 kDa), anti-FPN (69 kDa), anti-DMT1 (62 kDa), anti-hepcidin (2.8 kDa), anti-transferrin (TF, 79.5 kDa), and anti-ubiquitin (11 kDa) antibodies (as shown in Figure 3A). FTL and FTH showed right molecular weight (MW) bands suggesting that both FTL and FTH were present in basal ganglia, and levels were much higher in the control brain compared to PD or DS brain (Figure 3A). These results indicate that the presence of iron storage proteins (FTL and FTH) was higher in the control brains (Figure 3A,B, *p* < 0.0001). Whereas mitochondrial -ferritin (MtF) level was much lower than FTL and FTH that could be due to the heterogeneity of mitochondrial protein levels in the brain tissues and lowest in the DS brains (Figure 3A,B). Furthermore, the MtF levels in the PD brain compared to DS were not significantly different.

To investigate iron transport in the basal ganglia, FPN and DMT1 expressions were analysed in the same samples. There was low expression of FPN in the PD brains, and in DS subjects compared to controls (Figure 3C). The band intensity was analysed by IMAGE J and unpaired Student’s *t*-test and data indicated a statistically significant decrease of FPN in PD brains compared to controls (*R*^2^ = 0.69, *p* = 0.0001). These results indicate a defect in iron transport rather than iron storage in the PD and DS brain. The same samples were probed with anti-DMT1 (MW, 62 kDa) to further elucidate potential iron importer, and there was no overall change in DMT1 expression when comparing controls and PDs but higher in the DS brain (Figure 3C). To confirm any changes in the levels of iron regulatory protein hepcidin and transporter protein transferrin (TF) samples were examined using anti-hepcidin (MW, 2.8 kDa) and anti-transferrin (MW, 79.5 kDa) antibodies. Both proteins were significantly lower in PD and DS brains compared to controls (for hepcidin *p* = 0.0001, TF *p* = 0.001, Figure 3C). Hepcidin is a very small protein (MW, 2.8 kDa) and can easily cross the blood–brain barrier [26] and was found to be higher in PD compared to DS brains. As this could be due to defects in proteasomal degradation, we analysed the same samples using an antibody against ubiquitin protein and found that it was higher in the control samples but much lower in PD and DS (*p* = 0.01, Figure 3A,C). All samples were probed with an antibody against albumin (69 kDA) and β-actin (42 kDa) to adjust equal loading (Figure 3A).

### 2.5. Cellular Distribution of Ferritin Light-Heavy Chain and Mitochondrial Ferritin in Different Brain Compartments

Ferritin is an essential storage protein and is required for cellular proliferation from the embryonic stages to adult life in all living cells including brain, spinal cord, and muscles. To provide a detail expression pattern, mouse embryonic day 18 (ED 18) brain sections were stained with antibodies specific for ferritin (FTL, FTH, and MtF) and neuronal marker (nestin) and counterstained with 4′6-diamidino-2-phenylindole (DAPI) for nuclear staining, then imaged using a confocal microscope. FTL was more soluble and visible in the epithelial cells of the choroid plexus (CP), close to the blood vessels. FTH expression was visible in the CP outer membrane and that could be protecting the brain parenchyma from insults of excess haem iron or other metals. The MtF was visible in the neurons of cortical layers suggesting that all three ferritin proteins have a distinct role in the early stages of brain development (Figure 3D–F). FTH also expressed highly in the spinal cord (SC) (Figure 3G–I) and MtF was present in the heart muscles (Figure 3J,K), where energy demand for ATP production is very high. Heart muscle sections were also stained with mitochondrial markers, prohibitin and both proteins co-localised in the muscles (Figure 3J,K). All three ferritin protein levels decreased in brain disorders particularly in DS and very low levels of FTL were visible in the periphery of senile plaques (Figure 3L). In DS brains, extensive neuronal degeneration was observed, recognised by anti-Aβ42 antibody in the senile plaques and very low levels of anti-MtF staining, visible only in the surviving neurons (Figure 3M–O).

### 2.6. Ferritin Light Chain and Heavy Chain Proteins Localised in the Globus Pallidus and Putamen

To investigate cellular expression of iron proteins, brain sections of AD, PD, DS, and controls were analysed by immunohistochemistry (IHC) using DAB, and particular attention was given to the basal ganglia (Figure 4A). Associated nuclei of the basal ganglia reside in the diencephalon (subthalamic nucleus) and the mesencephalon (substantia nigra). As we have seen massive changes in the neuromelanin content in the SN of PD brain, we analysed ferritin protein expression in the SN, thalamus, globus pallidus, and putamen from AD, PD, DS, and age-matched controls. In control brains, SNpc revealed vast amount of granular neuromelanin that expressed ferritin particularly FTL, and found located very close to the blood vessels (Figure 4B), whereas in AD brain, FTL was present in the senile plaques, being particularly taken by the microglia (Figure 4C,D). Brain sections from basal ganglia (thalamus, globus pallidus, and putamen) from control and PD subjects when stained with monoclonal FTH, surprisingly, globus pallidus and putamen appeared loaded with small vesicles (Figure 4E,F) and on higher magnification it appeared like inclusions within the striosome-matrix compartments (Figure 4G–I). The inclusions were doughnut shaped with less staining in the middle, appearance very similar to Lewy body (Figure 4I–L). Interestingly, similar inclusions were seen in the basal ganglia of aged controls, and in DS brain (Figure 4J,K). Most of the vesicular inclusions were surrounded by lenticulostriate arteries, indicating copious blood supply to the area (Figure 4G–L). This finding suggests that FTH proteins may be stored in the striosome-matrix compartments (Figure 4E,F). We speculate that the ferroxidase property of FTH could protect different types of projection neurons (Figure 4G,J). For further investigation, another control brain section from the thalamus was stained with hepcidin and a very similar pattern of iron accumulation was seen, scattered throughout the thalamus (Figure 4M). These patterns are very different to the accumulation of hepcidin in the dentate gyrus where protein may enter from damaged blood vessels resulting from an inflammatory situation (Figure 4N) as reported previously [40]. The expression of hepcidin in the striatal white matter (internal capsule) and in the thalamus indicates that iron might be entering via the middle cerebral artery (MCA) or the lenticulostriate arteries (Figure 4O). Further investigations needed to confirm these findings.

### 2.7. Ferroportin Protein Expression Was Higher in the SNpc Than in the Hippocampus

Ferroportin is a protein transporting iron and other metals from endothelium at the basolateral site of blood vessels and could even be involved in transporting iron to the compartments of the basal ganglia via the lenticulostriate arteries [53]. To investigate cellular expression of ferroportin in the SNpc and basal ganglia, brain sections of PD, AD, DS, and controls were analysed by IHC using DAB staining. In controls, AD and DS brains, the neuromelanin cells and long axon and dendrites within SNpc were stained with ferroportin antibodies (Figure 5A–C). The neuromelanin cells were intact and punctate granular cells appeared stained on the periphery of the neurons. However, in the PD brains, most of the neurons were damaged and ferroportin accumulation was visible in the cell bodies and axons initial segments (AIS) (Figure 5D,E). In the hippocampus (HP) of control subjects, ferroportin expresses in the pyramidal neurons, with strong expression in the cell bodies and in the processes (Figure 5F), but less protein expression was seen in the HP of AD and DS brains (Figure 5G,H). Hepcidin was visible in the white matter, in the oligodendrocytes close to the blood vessels but with stronger presence in young DS brains compared to controls (Figure 5I,J). Furthermore, we stained a PD brain sections (from basal ganglia and close to blood vessels) and found evidence of hepcidin entering the brain parenchyma from damaged blood vessels (Figure 5K). We have stained SNpc and LC sections with DMT1 antibody. In control sections, DMT1 staining was visible in the neuromelanin cells, but less staining was observed in the LC with further reduction in the AD brain sections (Figure 5L,M). Both ferroportin and DMT1 proteins expressed in the wall of blood vessels and in the endosomes (Figure 5N,O). To identify the cellular types, such sections need to be stained with appropriate endothelial markers and analysed.

### 2.8. Cellular Localisation of Ferroportin and Hepcidin in the Cortex

For more precise co-localisation of hepcidin and FTL, brain sections from the cortex (HP and DG) were stained and analysed with confocal microscopy. In control brain, hepcidin (red) and FTL (green) were present in the HP and at higher magnification in the AD brain, microglia stained with FTL and hepcidin was located in the granule cell layer (Figure 6A,B). Most of the FTL positive microglia (green) were located in the vicinity of the blood vessels with some FTL protein entering the brain parenchyma from blood vessels shown with an arrow (Figure 6C). Control brain sections from HP and DG when stained with FTH, strong FTH staining in the pyramidal neurons were visible (Figure 6D). In the cortex of DS subjects, FTH was present in the senile plaques, whereas hepcidin was present in the peripheral cells (Figure 6E). Another AD brain section when stained with ferroportin and Aβ42, showed senile plaques visible in the cortex but ferroportin was present in the axon/dendrite filaments (Figure 6F). To evaluate any damage in the PD hippocampus, a PD brain (HP) section was stained with ferroportin and hepcidin and analysed through confocal microscopy. There were some damaged neurons with both proteins co-localised in the HP (Figure 6G–I). As we found FTL was visible in the microglia (Figure 6A–C), we assessed the structure of microglia using sections from control and DS brains stained with microglia marker (Iba1) and hepcidin. Ramified microglia (green) and hepcidin (red) were visible without co-localisation (Figure 6J), and in DS brain, activated microglia was present surrounding the plaques while hepcidin was seen in the surviving neurons (Figure 6K). Both hepcidin and ferroportin co-localised in the cortical neurons in DS brains (Figure 6L). These findings support the notion that all iron proteins expressed in the brain and are distributed in different locations.

### 2.9. Blood Vessel Damage and Basal Ganglia Pathology in PD, DS, and Ageing Brain

Damage to blood–brain barrier (BBB) and to small blood vessels are thought to be pathologically implicated in the basal ganglia disorders [54]. Small vessel disease (SVD) is initially seen in the arteries of the basal ganglia, mainly in the putamen and globus pallidus, as can be seen in Figure 5K. We stained basal ganglia sections from controls, DS, and PD subjects to evaluate this aspect further. In control blood vessels, DMT1 expressed in the endothelium of blood vessels and strings of GFAP positive astrocytes appeared probably to receive and transport the proteins into the brain parenchyma (Figure 6M), whereas in PD brain, the blood vessels appeared badly damaged with endothelium significantly losing its structure (Figure 6N). Another DS brain section with blood vessels (from basal ganglia) when stained with ferroportin and hepcidin, showed both proteins entering the brain parenchyma from the damaged blood vessel (Figure 6O).

Basal ganglia require very high energy, ATP, and many neurotransmitters including GABA and dopamine to maintain its functional activity. As the current study mainly focused on expression of iron proteins in the basal ganglia, we stained a striatum section with MtF and mitochondrial marker TOMM20 and then analysed with confocal microscopy (Figure 7A–C). The MtF showed immunoreactivity for the cell bodies and fibres of the caudate nucleus, and some co-localised with TOMM20 (Figure 7A–C). On higher magnification, MtF was visible in the neuronal fibres of striatal matrix (Figure 7C). One PD brain section was also stained with same antibodies, and both compartments of the striatum referred to as the striosomes (or patches) and extra striosomal matrix revealed co-localisation as well as staining of a neuronal cell body (most probably cholinergic neurons) and processes (Figure 7D–F). A control brain section from the SNpc (Figure 7J) and a PD brain section were stained with ferroportin and tyrosine hydroxylase (TH), a rate limiting enzyme for neurotransmitter dopamine. In control brain, dopamine cells were filled with TH proteins (red) (Figure 7J), whereas in PD brain, dopamine neurons were empty, but ferroportin staining was visible in the striatal matrix (Figure 7G–I). For further confirmation, two basal ganglia sections were stained dually with ferroportin and hepcidin or FTH and hepcidin. The staining indicated that ferroportin and hepcidin both expressed in the thalamo-striatal fibres in the DS brain (Figure 7K), where fibre bundles were positive for ferroportin. Brain sections stained with FTH and hepcidin showed patch-like terminal field of the thalamo-striatal projection in the putamen with co-localisation of both proteins (Figure 7L). These findings suggest that all iron storage and regulator proteins have distinct roles in the basal ganglia.

### 2.10. Expression of Iron Proteins in Primary Cultures of Hippocampal Neurons and Astrocytes

We have reported that ferroportin and hepcidin proteins are present in neurons and astrocytes and declines in Alzheimer’s disease brains [26]. To investigate the expression of ferroportin in vitro, we grew primary cultures of hippocampal neurons from embryonic day 18 rat brains and astrocyte cultures from postnatal day 3 rat brains. In serum free conditions, ferroportin protein was observed in cultured hippocampal neurons in the nucleus and perinuclear cytoplasm and maximum expression was observed in axons/dendrites (in the dendritic spines) and were photographed by confocal microscopy (Figure 8A–C). Hippocampal neurons were co-localised with GAD65 (a marker for GABA) (Figure 8C) or BIIIT a neuronal marker.

To examine iron expression in the astrocytes and other glial cells, we stained mixed glial cells with each ferritin antibodies and glial fibrillary proteins (GFAP) for astrocytes (Figure 8D–F). FTL was seen in the cell body and in the processes that co-localised with GFAP positive astrocytes (Figure 7D), whereas FTH showed prevalence in the oligodendrocyte like structures, although being GFAP positive (Figure 8E). Interestingly, a large proportion of mixed glial cells were MtF positive and GFAP negative, suggesting that these cells could be neurons, pre-mature bipolar cells not ready to choose its cellular fate (Figure 8F). Another mixed glia slide was stained with ferroportin and GFAP, some bipolar cells were ferroportin positive and GFAP negative, indicating that ferroportin expressed in all glia and neurons (Figure 8G). We further grew pure astrocytes in culture separating from mixed glia and stained with hepcidin and GFAP. Contrastingly, in astrocyte culture hepcidin expressed not only in the cell body but also extended into the processes (Figure 8H). The majority of the GFAP positive astrocytes showed extensive expression of hepcidin (Figure 8H,I). In astrocytes, hepcidin was present in the perinuclear region as well as in the growth cone (Figure 8H,I). These findings confirm that iron protein expresses in different glial cells and may have very specific roles.

## 3. Discussion

In diseases like AD, PD, and DS, brain iron homeostasis shows very different patterns in different brain compartments. Most dominant neuropathological changes observed in AD are senile plaques, which are deposits of amyloid β protein (Aβ42) in the brain parenchyma and blood vessel walls, along with cerebral amyloid angiopathy (CAA) and neurofibrillary pathology due to hyperphosphorylation of tau protein [28,55]. DS individuals with an additional copy of chromosome 21 develop full pathological features of AD by the time they reach middle age [26,30,31]. Both AD and DS display amyloid (Aβ42) pathology in the frontal and temporal cortex including hippocampus and dentate gyrus. Hence, studying brains from DS at different ages provides a ‘model system’ in which the earliest site of AD pathology can be observed and influences of other proteins linked to iron deposition can be monitored to assess its effects on disease progression.

Another debilitating neurological disorder is PD and is also linked with iron deposition in the brain. PD is pathologically characterised by the progressive degeneration and subsequent loss of dopaminergic neurons in the basal ganglia (caudate, putamen, SNpc, and LC). Although iron is unlikely to be the primary insult initiating the neurological conditions in AD and PD, it has been suggested that the accumulated iron has the potential to accelerate the progressive damage as iron storage protein (ferritin) is found in both senile plaques and in Lewy bodies [6,56].

Iron homeostasis in the adult brain shows profound differences when compared with the rest of the body due to the blood–brain barrier (BBB) and these do not reflect the changes seen in the serum [32,40]. However, with ageing, the brain becomes more susceptible and numerous factors (hypertension, abnormal glucose metabolism, and oxidative stress) could alter endothelium elasticity, leading to a leaky BBB. A leaky BBB is susceptible to facilitating the entry of many macromolecules and cytokine/chemokines into the brain parenchyma that can lead to neuroinflammation [57,58,59].

Over the last two decades, many groups have thoroughly studied the role of ferritin [3,14,60,61] TF and the TFR1 [10,39] in the liver, erythrocytes, and other major organs including the brain [43]. However, the question as to how iron enters the brain parenchyma and whether it depends on TF and/or DMT1 remains unanswered.

Iron accumulation increases with age in human brain. Iron, stored as a ferritin, and all the three types of ferritin (heavy chain, light chain, and mitochondrial) may play different roles in different cells [62]. Ferritin sequesters and stores iron, consequently protecting cells against iron-mediated free radical damage. However, the mechanism behind iron exit from the ferritin cage and its re-utilisation in the brain are largely unknown. In the current study, we demonstrate that all three isoforms of ferritin (FTL, FTH, and MtF) are present in the brain and are essential for early brain development (Figure 3D–F). FTL is mainly found in the microglia and soluble protein may enter from blood vessels and be taken up by end feet of astrocytes as reported previously [26], whereas in healthy brain, FTH is stored in the oligodendrocytes and astrocytes. In this study, however, we focused on assessing the expression of mitochondrial ferritin (MtF) in normal and disease brains; on higher magnification, MtF was visible in the neuronal fibres of striatal matrix. Mitochondria plays a very important role in the muscle and in the brain, aiding production of ATP and many different neurotransmitters [12].

Ferritin is often associated with iron deficiency, hypoferritinemia, hypoxia, and immune complications, and are all significant concerns for systemic infections in Alzheimer’s disease and Down syndrome dementia [40,63]. As iron accumulates with age in the human brain, an increase in the total iron level (as ferric iron) was found in neuromelanin containing neurons of SNpc of PD [42,45,64]. We have shown by biochemical analysis of SNpc and LC that non-haem iron levels were much higher in PD compared to, AD and older controls (Figure 1A,B). In young control subjects (YC) and in young DS subjects, the iron level was significantly lower compared to that in older subjects (Figure 1A). These findings suggest that iron accumulation in neuromelanin, particularly in the SNpc is age-related and even could be disease related like in PD.

We further examined mRNA expression of ferritin (FTL), ferroportin, and DMT1 in human AD, DS, PD, and age-matched control brain tissues by in situ hybridisation. Ferroportin mRNA expression was very limited in PD and DMT1 and ferroportin mRNA and both proteins (ferroportin and DMT1) were found to express in the axon initial segments (AIS) of neuromelanin cells, suggesting proteins to function as solute carriers (Figure 2). The AIS is involved in exchanging sodium and calcium ions. It was also reported that ferroportin is involved in transporting calcium and other metals [65]. In this study, we report for the first time that AIS may be implicated in metal ion exchange in certain selective neurons.

Both FTL and FTH are present in the neuromelanin cells, whereas DMT1 and ferroportin are seen in the neuromelanin cell membrane. However, the majority of the neuromelanin cells in the PD brains were damaged and the contents of neuromelanin vesicles were scattered throughout the SNpc. Some of the vesicles may contain haem-bound iron products such as haemosiderin and lipofuscin and engulfed by microglia, or in some cases engulfed by endosomes. These molecules, however, are highly auto-fluorescent complicating their analysis. We speculate that glial cells (microglia and/or astrocytes) failed to clear such lipofuscins or haemosiderin molecules and perish, aiding in the formation of Lewy bodies. The Lewy bodies were positive for FTL, FTH, and α-synuclein. The Lewy bodies were found in the early stages of PD (1–3) but in late stages (4–6) there were hardly any intact dopaminergic neurons visible except Lewy neurites and lipofuscin molecules visible as described previously [66]. Our data complemented previous findings from other groups [39,42,43]. Increased L-ferritin was seen in the neuromelanin cells of control brains but less in the SNpc of PD subjects, and some punctate ferritin was observed within the Lewy bodies. Whether the brain requires L-ferritin to store iron or dopamine waste products are bound to L-ferritin remains to be elucidated.

Neuromelanin is a complex-molecule located in the double-membrane organelles, particularly found in the dopaminergic neurons of the SN and noradrenergic neurons of the locus coeruleus (LC) situated in the reticular substance of the upper pons [67,68,69]. The LC is the major source of noradrenaline modulation in the brain and has been shown to be involved in regulating a wide range of higher cognitive functions, such as working memory, learning and attention, and deficits in noradrenaline may contribute to the pathophysiology and symptomatology of several neurological disorders including AD and DS [70,71,72]. Neuromelanin contents were comparatively higher in the LC of young DS subjects than young controls. Neuromelanin cells can also chelate other redox active metals (Cu, Mn, and Cr) and toxic metals (Cd, Hg, and Pb), buffering their toxic potentials [73,74]. We would like to speculate that the clearing processes of iron from LC is much more efficient than substantia nigra.

Basal ganglia comprise of subcortical structures primarily involved in motor control and motor learning. We have concentrated our evaluation of iron homeostasis in the basal ganglia of PD and DS brains using Western blotting (WB). Our analyses support the argument that there are more FTH proteins found in the caudate and putamen. The histochemical analysis revealed very strong FTH protein expression in the striosomes and matrix compartment, in the white matter fibres of internal capsule (Figure 4E), and putamen appeared loaded with small vesicles (Figure 4E,F). On higher magnification, these vesicles appeared like Lewy bodies within the striosome-matrix compartments, appearing doughnut-shaped with a less central staining (Figure 4I–K). There are many blood vessels present in the basal ganglia, middle cerebral artery (MCA, a continuation of the internal carotid artery), lenticulostriate arteries and medial striate artery (of Heubner), which could carry FTH with TF bound diferric iron and release via endosomes in the basal ganglia. The astrocytes in close proximity of the blood vessels seem to be the main iron carrier and are capable of storing large amounts of iron in the form of FTH in the striosomes, and most of the vesicular inclusions were surrounded by lenticulostriate arteries, indicating copious blood supply to the area. This finding indicates that FTH protein may be stored in the striosome-matrix compartments. We would like to suggest that the ferroxidase property of FTH could be to protect the different types of projection neurons by providing adequate nutrients.

Damage to the blood–brain barrier and small blood vessels is assumed to be pathological and linked to basal ganglia disorders [75]. DMT1 protein expression was seen in the lining of the endothelial cells of blood vessels in the apical site and could be involved in importing ferritin and hepcidin through the end-feet processes of astrocytes (Figure 5N,O). There were no differences in DMT1 protein expression observed in the PD or control brains.

Much higher levels of ferroportin were visible in the SNpc of PD brains compared to AD or DS brains. Both ferroportin and DMT1 proteins express in the epithelial cells of the choroid plexus and act like a gate keeper between the brain and ventricles (Figure 6O) [26,76]. Ferroportin expression is heterogeneously reduced with age in controls and severely reduced in AD brains as we have previously reported [26]. Similarly, we could show here that ferroportin levels are also reduced in the DS brains as a result of neuronal cell loss.

We had previously reported that ferroportin and hepcidin were present in neurons and astrocytes and their levels declined in AD brain [26]. We found significant correlation between hypoferritinemia and elevated levels of serum hepcidin in AD and DS [40,63]. In normal brain, hepcidin is present in the hippocampus and in the granule cells of dentate gyrus, whereas FTL is visible in the microglia and is involved primarily in the pruning and clearing process (Figure 6). In PD brains, hepcidin and ferroportin are present in the hippocampus and are co-localised. Similarly, in DS brains, hepcidin is present close to the neurons and may have a protective function, and aided by ferroportin, reducing further damage through internalisation and digestion of the dead cells. These findings indicate that in the early stages, particularly in young DS subjects, hepcidin may have an important role in sequestration and utilisation of iron. Hepcidin can be transported via the macrophages and the majority of the vesicular hepcidin enters the brain via a compromised blood–brain barrier [77]. Indeed, we observed that in close proximity of the blood vessel, hepcidin was visible in the erythrocytes in the PD and DS brains (Figure 6N,O), indicating that it possibly arrives through a leaky blood–brain barrier before binding and internalising ferroportin as previously described [40].

The anatomical connections between globus pallidus and putamen form the important elements of the sensory, motor, cognitive, and motivational networks of the brain [53]. When active, these connections require very high energy, ATP, and many neurotransmitters including GABA and dopamine. In the current study, we found MtF present in the neuronal cell body (most probably representing cholinergic neurons) and processes, additionally, in fibres of the internal capsule and extra striosomal matrix where both proteins co-localised. Our observations suggest that MtF has a role in the production of neurotransmitter and/or in maintaining high energy activity. MtF expression, however, was limited in the AD and DS brain.

The substantia nigra is the source of dopaminergic modulation involving the basal ganglia. These neurons contain tyrosine hydroxylase and the dopamine precursor neuromelanin. In control brains (especially in substantia nigra) strong expression of ferroportin and the presence of TH in dopaminergic neuron were observed (Figure 7J), while in the PD brain sections, ferroportin expression was very faint without any TH staining (Figure 7G–I).

Ferroportin is present in normal neurons, but is lost in PD brains. Ferroportin expressed in the pre-synaptic vesicles indicate that it is a vesicular transporter protein carrying α-synuclein and iron to synaptic terminals. It is also an important retrograde solute carrier, not only binding iron, but also other metals (calcium, manganese, copper, cadmium, and aluminum) [78,79]. In human brain, ferroportin is expressed in medium spiny neurons (SMNs) of the striatum, mainly containing γ-aminobutric acid (GABA), and in dopaminergic neurons containing melanin pigments. In PD brains, ferroportin loss could therefore lead to impaired iron homeostasis in the motor neurons. However, it remains unclear whether increases in iron are due to a failure of iron usage (excessive iron storage), or a failure of the clearance processes (transport defect). Both DMT1 and ferroportin expression were irregular in PD brains, whereas ferritin, transferrin, and ubiquitin expressions were decreased compared to controls as confirmed by WB. Conversely, α-synuclein expression was consistent in control samples and higher in PD brains. It was reported that the *5′-UTR* of α-synuclein gene contains a putative iron responsive element [79]. Therefore, increase in iron may lead to a post-transcriptional upregulation of α-synuclein protein [80].

The ubiquitin expression was lower in PD samples, suggesting a defect in the ubiquitin–proteasome pathway. Many groups have described defects in this pathway (parkin, Pink1, LRRK2 protein) and autophagy in PD [81,82]. However, it is still not clear, which iron proteins are involved in brain iron degradation processes, but our data indicates the strong possibility of involvement of hepcidin, that was mainly located in the lysosomal vesicles in PD brains, and some co-localised with LAMP1 protein suggesting the involvement of hepcidin in lysosomal degradation.

Our hippocampal cell culture shows that ferroportin is expressed in the neurons and astrocytes. It was observed in cultured hippocampal neurons, especially in the nucleus and perinuclear cytoplasm, and maximum expression was observed in the axons/dendrites (in the dendritic spines) alongside co-localising with GAD65 (a marker for GABA) or BIIIT (a neuronal marker). All three ferritin isoforms (FTL, FTH, and MtF) were expressed in the astroglia. In contrast to that, in astrocyte culture, hepcidin not only expressed in the cell body but expression also extended to the processes and growth cones. These findings confirm that iron proteins express in different glial cells and may have a specific functional role in developing neurons (in growth cone formation) and in the glia, including astrocytes and oligodendrocytes.

Several groups recently reported trans-synaptic propagation of α-synuclein protein in mouse and rat models [83,84]. In the rat model, authors proposed bidirectional α-synuclein propagation via the vagus nerve, i.e., duodenum-to-brainstem-to-stomach [83]. However, we have not seen any synuclein-like protein propagation from either heart or skeletal muscle to the brain parenchyma in control or disease brains. Animal models are very useful to investigate functional characteristics, but they cannot recapitulate all pathological and clinical features observed in human neurodegenerative diseases. Similarly, another paper investigated tau spread by expressing human tau via viral vector in old and young mice. The old mice showed increased tau spreading in the hippocampus and the conclusion was that age-related brain region specific tau spread could be a related risk for sporadic AD [85].

## 4. Materials and Methods

### 4.1. Ethics and Participants

National Research Ethics Committee of the East of England, Cambridgeshire, UK granted approval for the study. Cambridge Health Authorities Joint Ethics Committee granted ethical approval for the use of human brain tissue from PD and age-matched controls (REC ref 02/193) and for Alzheimer’s and Down syndrome (project ref no. REC:15/WM/0379). Information for AD, DS, and controls has already been disclosed in previous publications [26,40].

### 4.2. Human Brain Tissue

Human paraffin embedded brain tissue sections from basal ganglia (BG), including substantia nigra (SN), from 6 cases of PD (age 66–78 years, mean age 72.5 years), 6 older controls (subjects who had died without neurological complications, age 73-88 years, mean age 82.3 years) with similar post-mortem intervals (20–81 h) were obtained the from the UK Brain Bank. Only one case (PD5: 0975) had long post-mortem delay (125.30 h), and was younger (66 years) than rest of the PD cases (see Table 1). Human post-mortem brain tissues from younger controls (mean age 60 ± 15 years) and from people with DS (mean age 60 ± 15 years) (*n* = 6 in each group) were provided by the Cambridge Brain Bank (Table 1). All methods have been described in our previous publications. Fresh unfixed tissue was frozen at autopsy in isopentane chilled with liquid nitrogen and subsequently sectioned with a cryostat (15 μm thickness) and stored at –80 °C until analysis. All work carried out was approved by the Ethical Committee of Addenbrooke’s Hospital, University of Cambridge, UK.

### 4.3. Animals

Three-month-old C57/Bl6 mice and three-month-old Sprague-Dawley (SD) rats were purchased from Charles River and bred for the experiments at University of Cambridge Bioscience facility (UBSS, at John van Geest Centre for Brain Repair animal facility, Cambridge, UK). All experiments were performed according to the “Animal (Scientific Procedures) Act 1986” and the “Guidance of the Operation of ASPA 2014”, and were approved by the Ethical Committee of the UK Home Office.

### 4.4. Total Non-Haem Iron Determination

Mid brain samples were obtained during autopsies of male and female subjects who died at different ages (as shown in Table 1). Samples of SN and LC were carefully dissected and stored at –80 °C until use. The microanalysis of non-haem iron content of brain tissues was determined by the modified methods of Torrance and Bothwell [52]. Brain tissue from AD, PD, DS, and age-matched controls (25 mg of tissues from SN or LC of each subject, *n* = 4) homogenates were prepared 1:10 (W/V) using high-purity water, using FastPrep-24 homogeniser with Garnet Matrix 2 mL tubes (MP, Fisher Scientific, Pittsburgh, PA, USA). Tissues were weighed and digested for 48 h in 10% trichloroacetic acid/10% HCl at 65 °C. Two hundred microliters of the extract were then added to 1 mL of chromogen solution (0.01% bathophenanthroline-disulfonic acid, 0.1% thioglycolic acid, 7M sodium acetate) and incubated for 10 min, and the absorbance was measured at 535 nm in a DU650 spectrophotometer (Beckman Instruments, Fullerton, CA, USA). A certified iron standard from Sigma (kit 565A, Sigma, Burlington, MA, USA) was used to determine iron levels. A standard curve performed for iron concentrations between 10 and 500 µg/mL revealed a linearity of response with a slope of ~1. Samples were diluted appropriately to fall within the linear range, and a 100 µM FeCl_3_ solution was used as an additional internal control. TNHI values were expressed as micrograms of iron per gram of wet weight.

### 4.5. Perls’ Staining Methods for Brain Iron in the NM Cells

Perls’ Prussian blue method was followed for staining ferric iron: The ferric iron combines with potassium Ferrocyanide to form the insoluble Prussian blue precipitate is as follows:4FeCl_3_ + 3K_4_Fe(CN)_6_ = Fe_4_[Fe(CN)_6_]_3_ + 12KCl       (ferric iron) (potassium (ferric ferrocyanide) ferrocyanide)

Paraffin fixed brain sections (*n* = 6, in glass slides) were deparaffinised and hydrated with iron free distilled water. The sections were then transferred to a mixture of equal parts of 2% potassium ferrocyanide at pH6 and 2% HCL in distilled water for 20–30 min. The brain sections were then washed in distilled water. All sections were counterstained in eosin, dehydrated, and mounted with a synthetic resin medium [63].

### 4.6. In Situ Hybridisation

Brain tissues were prepared for in situ and probed as described previously [77]. Briefly, oligonucleotide probes corresponding to human Ferroportin (FPN-Hu 5′ccagaaacacagacaccgcaaagtgccacatccgatc), DMT1 (DMT1-Hu-5′gacagctcagcctgaactctatcttctgaacaccatggac), and ferritin (FTL-Hu-5′ttatcaagaagaatgggtga ccaacctgaccaaactncacaggc) were labelled with [^35^S]-dATP (Dupont-NEN) using a 3′-terminal deoxynucleotidyl transferase enzyme kit (Boehringer-Mannheim), and hybridisation was performed in a humid chamber overnight at 60 °C (~16 h). Sections were then washed, dehydrated, and air-dried before being exposed to Biomax MR film (Kodak) for 14 days. Non-specific hybridisation was abolished in the presence of 100-fold excess unlabelled oligonucleotide. To facilitate the differentiation at a cellular level, sections were dipped in Ilford K-5 emulsion (Ilford, UK), stored at 40 °C in light proof boxes for 56 days, developed in phenisol (Ilford), fixed, and counterstained in 0.2% methylene blue.

### 4.7. Antibodies

The following antibodies were used: monoclonal anti α-synuclein (LB 508, Zymed, San Francisco, CA, USA, 1:2000 for Western blotting (WB), 1:1000 for immunohistochemistry (IHC) or immunofluorescence (IF), monoclonal anti tyrosine hydroxylase (Chemicon, Richmond, UK 1:1000 for IF), monoclonal anti-glial fibrillary protein (GFAP, Sigma, UK, 1:2000, for IF), monoclonal anti-Iba1 (Thermo Fisher, UK, MAB M1/70), polyclonal anti-Iba1(Wako cat number 019-19741, USA), monoclonal anti β-actin (Sigma, UK, 1:5000 for WB, 1:2000 for IH), monoclonal anti-transferrin (Abcam, Cambridge, UK, 1:1000 for WB, 1:500 IF), monoclonal ferritin light chain (Abcam, ab69090,UK, 1:1000 for IF), polyclonal ferritin heavy chain (Abcam, ab65080, UK, 1:500 for WB, 1:1000 for IF), polyclonal mitochondrial ferritin (Abcam, ab93428, UK, 1:1000 for IF, 1:500 for WB), rabbit polyclonal anti-ferroportin (Abcam ab85370, UK 1:500 for WB, 1:200 for IHC), monoclonal ferroportin (Abcam ab93438, UK, 1:1000 for IF), rabbit polyclonal anti-DMT1 (Abcam, ab55812, UK 1:200 for WB, 1:250 for IHC), polyclonal anti-hepcidin (Abcam, ab30760, UK, 1:200 for IF, 1:100 for WB), rabbit polyclonal anti ubiquitin (DAKO, 1:500 for WB). The following secondary antibodies were used: biotinylated goat anti-rabbit and biotinylated horse anti-mouse (both from Vector Laboratories, 1:250 for IHC); Alexa Fluor 568-labelled donkey anti-mouse, Alexa Fluor 488-labelled donkey anti-rabbit, and Alexa Fluor 568-labelled donkey anti-goat (all from Invitrogen, Pittsburgh, PA, USA, 1:1000 for IF).

### 4.8. SDS-PAGE and Western Blotting

Protein lysates were prepared from basal ganglia and SNpc of human brains from PD, DS, and age-matched controls (*n* = 6 in each group) as described previously [26]. 20 μg protein samples were separated on 15–20% Bis-tris gels and transferred to 0.45 or 0.2 μm pore size PVDF membranes (Invitrogen). The membranes were incubated with the appropriate primary antibody in blocking buffer for 24 hour at 4 °C and then washed three times with 0.1M tris saline buffer with 1% Tween 20 (TBST) followed by incubation for 1 h at room temperature (RT) with HRP-conjugated secondary antibodies (anti mouse IgG 1:3000, DAKO, Carpinteria, CA, USA) or anti-rabbit IgG (1:3000; DAKO, USA) antibodies. Binding was detected with ECL Plus chemiluminescence reagents and Hyperfilm ECL (both from GE Healthcare, Chicago, IL, USA).

### 4.9. Immunohistochemistry

Paraformaldehyde (PFA) fixed tissues were first quenched with 5% hydrogen peroxide and 20% methanol in 0.01 M PBS for 30 min at room temperature (RT) followed by three rinses for 10 min in 0.01 M phosphate buffer saline (PBS). Non-specific binding sites were blocked using blocking buffer (0.1 M PBS, 0.3% Triton-X100, and 10% normal goat serum for polyclonal antibodies or 10% normal horse serum for monoclonal antibodies) for 1 h at RT. Tissue sections were incubated overnight with the primary antibody diluted in blocking buffer. Binding of the primary antibody was detected using a biotinylated secondary antibody goat anti-mouse IgM (Jackson ImmunoResearch, West Grove, PA, USA) followed by an avidin-biotin horseradish peroxidase complex (Vector Laboratories, Burlingame, CA, USA). Slides were developed with diaminobenzidine substrate (DAB) with nickel enhancement (Vector Laboratories, USA), mounted and photographed

### 4.10. Immunofluorescence (IF)

Brain sections were blocked using blocking buffer (0.1 M PBS, 0.3% Triton X100, 10% normal donkey serum) for 1 h at RT, then incubated overnight at 4 °C with primary antibody diluted in blocking buffer. Alexa Fluor-conjugated secondary antibodies were used for detection and samples counterstained with 4′6-diamidino-2-phenylindole (DAPI, Sigma). Sections were then mounted on glass slides with coverslips using Fluoro Save (Calbiochem).

### 4.11. Primary Rat Hippocampal Cultures

Cultures of dissociated hippocampal neurons were prepared using previously described methods with some modifications. Hippocampal tissue was collected from Sprague-Dawley (SD) rat embryos (Charles River, Hollister, CA, USA) at day 18 of development, digested in 0.25% trypsin for 15 min at 37 °C in Hank’s buffered saline solution (HBSS) and then gently triturated in Dulbecco’s modified Eagle’s medium (DMEM) supplemented with 10% foetal calf serum and an additional 0.8% of glucose. This produced a single cell suspension that was plated at a density of 5 × 10^4^ cells/cm^2^ on glass coverslips coated with 250 µg/mL of poly-D-lysine (Sigma, USA). Cultures were maintained in Neurobasal medium supplemented with 2% B27 and 1% GlutaMAX, and used in experiments after 3–4 weeks in culture. All cell culture media and reagents were purchased from Invitrogen, UK. After 1–3 days, the culture medium in each well was changed every third day. Confluent cells were fixed with 4% PFA and washed with PBS. Coverslips were analysed by IF staining as above and examined by confocal microscopy.

### 4.12. Primary Rat Astrocyte Culture

The cortices of new-born rats (SD, postnatal P1–P3 days old) were removed, stripped of meninges, chopped up with a razor blade, and incubated in 0.25% trypsin (Sigma) in EDTA for 15 min at 37 °C. Following a 5 min incubation in DNAse (0.001%, Sigma in HBSS), the supernatant was removed and tissue triturated in 2 mL of triturating solution with a flame-polished Pasteur pipette. After centrifugation for 3 min at 1000 *g*, cells were re-suspended in culture medium (10% foetal calf serum, Sera Lab, Sherman Oaks, CA, USA; 1% penicillin–streptomycin–fungizone (PSF), Gibco, in Dulbecco’s modified Eagle’s medium (DMEM), Gibco) and plated in 75 cm^3^ poly- l-lysine-coated tissue culture flasks (Nunclon, Life Technologies, Paisley, Scotland) at a density of 2 cortices per flask. To prepare pure astrocytes culture, after cells became confluent, mixed glial cultures were agitated on a rotary shaker at 200 r.p.m. for 24 h. Cultures were washed to remove floating cells and treated with l-leucyl-methyl-ester (10 mm) for 1 h to kill any remaining microglia. Astrocytes were removed with trypsin and plated on 13-mm-diameter poly-D-lysine-coated coverslips at a density of 10^4^ per coverslip. After 1–3 days, the culture medium in each well was changed and then washed with PBS and fixed as described above.

### 4.13. Microscopy

Bright field images were taken and quantified using Lucia imaging software and a Leica FW 4000 upright microscope equipped with SPOT digital camera. Fluorescence images were obtained using a Leica DM6000 wide field fluorescence microscope equipped with a Leica FX350 camera and 20× and 40× objectives. Images were taken through several z-sections and deconvolved using Leica software. A Leica TCS SP2 confocal laser-scanning microscope was used to acquire high-resolution images.

### 4.14. Image Analysis and Statistics

IHC and Western blot images were quantified using ImageJ software (US National Institutes of Health). Data were analysed by paired Student’s *t*-test (two-tailed) for two group comparison, or by ANOVA test for multiple comparison testing. Values in the figures are expressed as mean ± SEM. A one-way ANOVA was used for comparison of data among control, PD, AD, and DS and conducted with IBM-SPSS statistic 19 software. Significance was analysed using GraphPad and *p*-values ≤ 0.001 were considered significant and are indicated in the corresponding figures and figure legends.

## 5. Conclusions

Many iron proteins participate in brain iron homeostasis. The three ferritins (heavy chain, light chain, and mitochondrial ferritin) may have different roles in different cells. Ferroportin is a key player to transport iron in the brain and DMT1 may have a supportive role in importing iron through astrocytes. The finding that hepcidin and ferroportin were co-localised in cortical neurons in control brains is consistent with the role of these proteins in regulating neuronal iron release. Without ferroportin, hepcidin is unable to participate in degradation and ubiquitination processes. This imbalance in brain iron homeostasis may lead to oxidative stress in cells, finally causing cell death.

In conclusion, our work indicates that PD is a complicated and multifactorial disorder with varying brain pathology when compared to AD or DS dementia. The pathological changes affecting PD mainly involve the nigrostriatal pathways, whereas direct evidence of iron mishandling in AD and DS brain comes from the histochemical demonstration of non-haem iron deposits found in senile plaques. However, all three diseases pathognomically show extensive endothelial dysfunction and damage of the blood–brain barrier as well as brain ischaemia [86]. Loss of vascular integrity is also responsible for abnormal iron accumulation in the form of haem-positive granular deposits in addition to ferritin in AD brains. These have been previously demonstrated in aged brains in association with senile plaques and results from capillary bleeds or micro-haemorrhages [87]. Impairment of iron homeostasis could be a consequence of several other factors such as ageing, mitochondrial dysfunction, oxidative stress, and protein aggregation. Our data do support the hypothesis of age-related risk for sporadic AD, as we have noticed a strong link between age-associated increase in iron stores in the brain alongside a failure of clearing processes, and an increasing incidence of AD in DS subjects with advancing age.

## Figures and Tables

**Figure 1 ijms-23-01060-f001:**
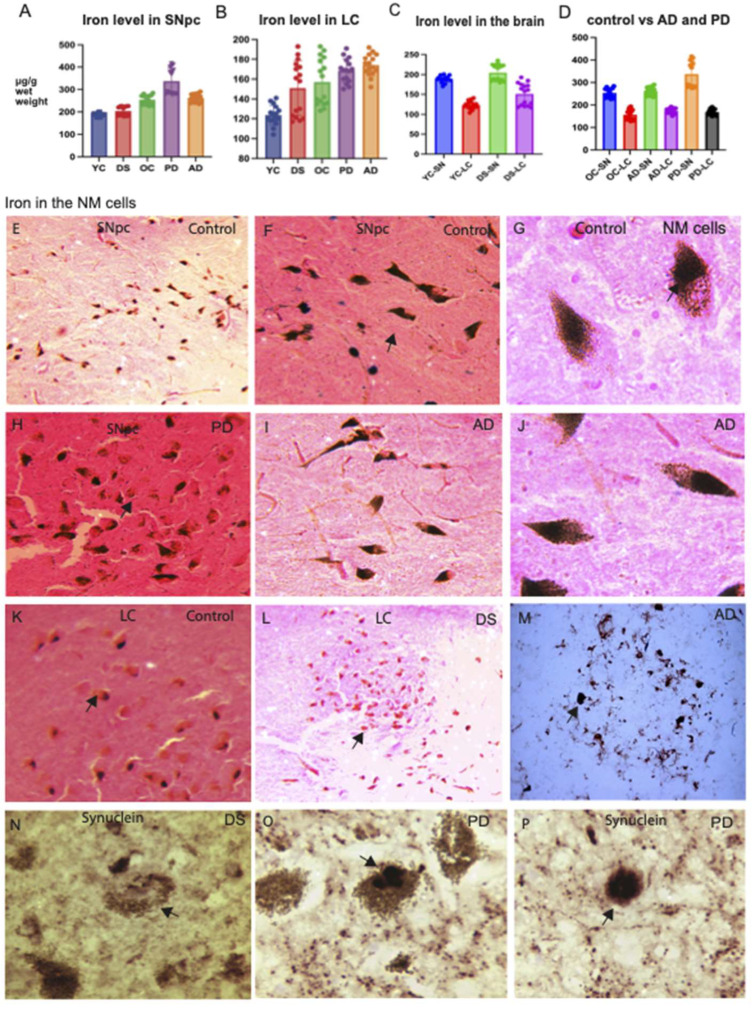
Non-haem iron concentration and neuromelanin expression in controls vs. disease brains. The non-haem iron concentration (NHIC) was measured in the SNpc and LC of AD, PD, DS, young control (YC), and old control (OC) subjects. The highest iron levels were found in the SNpc of PD brains (**A**). The concentration of iron in LC was significantly lower in all four groups compared to SNpc (**B**). Iron levels in SNpc and LC were compared in YC versus DS subjects (**C**) and in older controls (OC) versus AD and PD subjects (**D**). The concentration of iron in SNpc increases with age unlike that seen in LC (*R*^2^ = 0.76, *p* < 0.0001). SNpc and LC brain sections of AD, PD, and DS, and age-matched controls were stained with Pearls’ stain and in the control subjects’ SNpc, black granular iron molecules were visible in the neuromelanin (NM) containing cells (**E**–**G**). In higher magnification, expression was visible in the damaged NM cells of the PD (**H**), whereas intact NM cells with processes were visible in the AD brains without any cell damage (**I**,**J**). Neuromelanin content in the LC neurons in control brains was low (**K**), and even lower in the DS subjects (**L**). Iron accumulation was seen in senile plaques in AD brain (**M**) and α-synuclein protein expression was located in the neuromelanin cells in the DS and in PD brains (**N**–**P**). Lewy bodies were present inside the NM cells (**O**,**P**). Scale bar: (**E**,**L**,**K**) = 100 μm, (**F**,**H**,**I**) and **M** = 50 μm, (**G**,**J**,**N**–**P**) = 20 μm.

**Figure 2 ijms-23-01060-f002:**
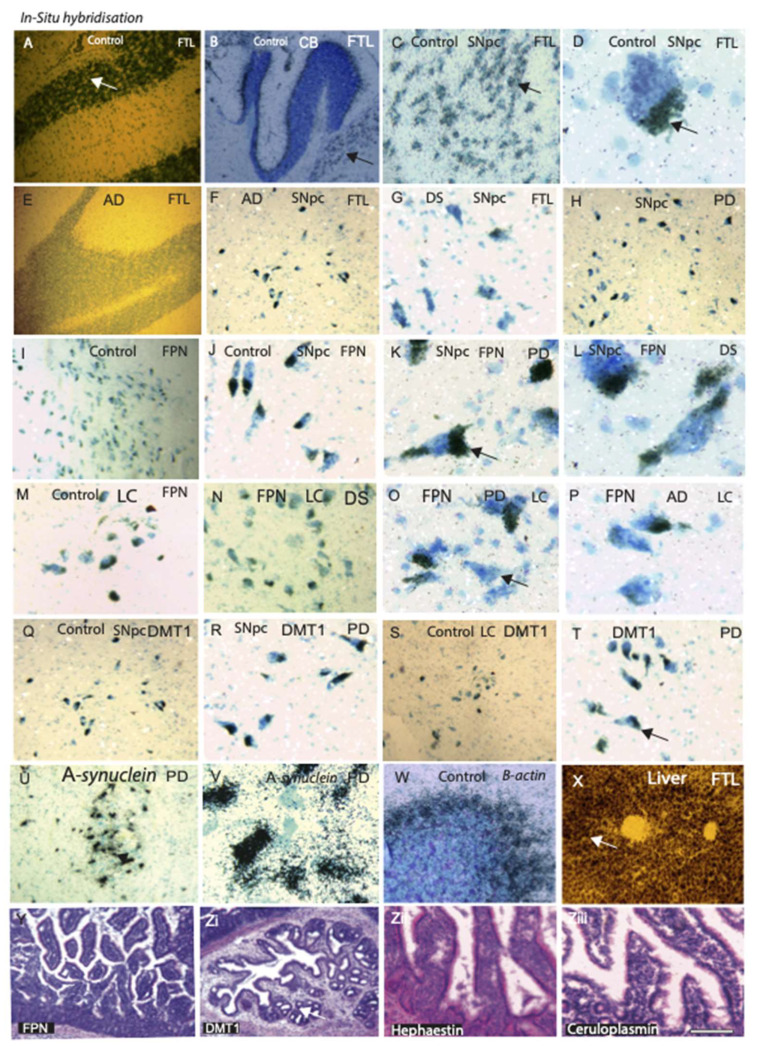
FTL, FPN, and DMT1 mRNA expression in AD, PD, and DS brain tissues. The mRNA expression of FTL, FPN, and DMT1 were detected by in situ hybridisation of AD, DS, PD, and age-matched control brain tissues. In control brain sections, ferritin (FTL) mRNA was visible in hippocampus (HP) and dentate gyrus granule cells (**A**) and cerebellum (**B**). In the SNpc, NM cells showed strong hybridisation signal of FTL (**C**,**D**), indicated by arrow) and minimal signals in the HP of AD subjects (**E**). FTL mRNA was visible in the PD and AD SNpc (**F**,**H**), whereas the DS brain (SNpc) showed very low levels of FTL mRNA expression, and most of the NM cells were empty (**G**). The ferroportin (FPN) was visible in the NM cell membrane of control brain sections (**I**); higher magnification (**L**), whereas it was limited in PD (**K**) and DS (**L**) brains and was only detected in the axon initial segment (AIS) of the NM cells (**J**–**L**). There was less ferroportin mRNA expression in LC when compared to SNpc in control (**M**), DS (**N**), PD (**O**), and AD (**P**) brain. DMT1 mRNA was detected in the AIS of the NM cells in SNpc in control (**Q**), in PD (**R**), and in LC of control (**S**) and PD brain, indicated by arrow (**T**). Two PD brain samples were hybridised with α-synuclein probes, which was highly expressed in the SNpc of PD brain (**U**,**V**). A β-actin probe was showing strong expression in the cerebellum (**W**). A mouse liver section was probed with FTL showing strong signal (**X**). Ferroportin and DMT1 mRNA was expressed mainly in the gut endothelium, and when mouse gut sections were probed with ferroportin (**Y**), DMT1, hephaestin, and ceruloplasmin (**Zi**–**Ziii**), results showed mRNA expression in the gut mucosa. Scale bar: (**A**–**C**,**E**–**J**,**M**–**Q**,**W**,**X**) = 100 μm; (**D**,**J**–**L**,**O**,**P**,**V**) = 25 μm; (**Y**,**Zi**–**Ziii**) = 20 μm.

**Figure 3 ijms-23-01060-f003:**
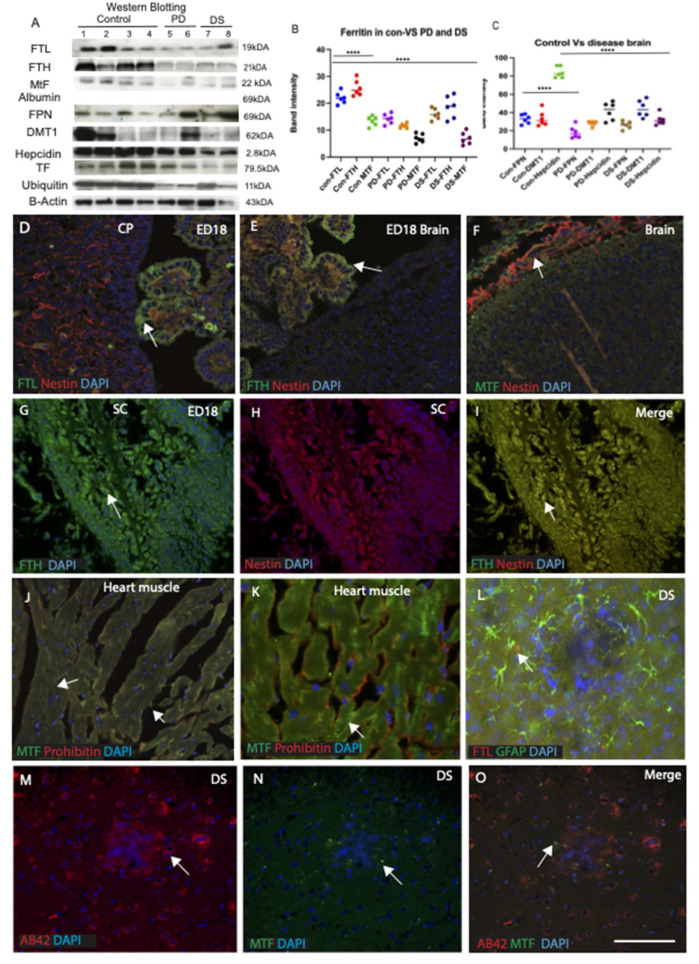
Ferritin light-heavy chain and mitochondrial ferritin protein levels and cellular distribution in different brain compartments. Western blotting (WB) of basal ganglia brain was analysed using PD, DS, and age-matched control samples (*n* = 6). A single ~19 kDa band consistent with FTL, FTH (21 kDa), MtF (~22 kDa), hepcidin (~2.8 kDa), TF (79.5 kDa), and ubiquitin (11 kDa) were detected (**A**). The highest levels were seen in the control brains (lane 1–4) compared to PD (lane 5,6) and DS subjects (lane 7,8) (**A**). Densitometric analysis of the blots showed significant differences between control, PD and DS brains (**A**,**B**), *p* < 0.0001). In contrast, mito-ferritin (MtF) level was much lower in the PD compared to controls and the lowest in the DS brains (**A**,**B**). There was low expression of ferroportin in PD brains compared to controls and DS subjects (**C**). Albumin and β-actin loading control was used to normalise data (**A**). The band intensity was analysed by IMAGE J and unpaired Student’s *t*-test and data indicated a statistically significant decrease of ferroportin in PD brains compared to control brains (*R*^2^ = 0.69, *p* = 0.0001), ******** = *p* < 0.0001. Mouse embryonic day 18 (ED 18) brain sections were stained by double immunofluorescence with ferritin (FTL, FTH, and MtF) and neuronal marker (Nestin) and counterstained with DAPI for nuclei (blue), then imaged using a confocal microscope. FTL was visible in the epithelial cells of the choroid plexus (CP) and close to the blood vessels (**D**). FTH expression was visible in the CP outer membrane (**E**) and MtF was visible in the neurons of the cortical layers (**F**). FTH expressed highly in the spinal cord (**G**–**I**). The heart muscles were stained with MtF and prohibitin (mitochondrial markers), both proteins co-localised in the muscle cells (**J**–**K**). In DS cortex, very low levels of FTL were visible in the periphery of senile plaques (**L**). DS brain section stained with anti-MtF and anti-Aβ42 showed that MtF staining was only visible in the surviving neurons distributed around the periphery of amyloid plaques (**M**–**O**). Scale bar: (**D**–**K**) = 20 μm, (**L**–**O**) =15 μm.

**Figure 4 ijms-23-01060-f004:**
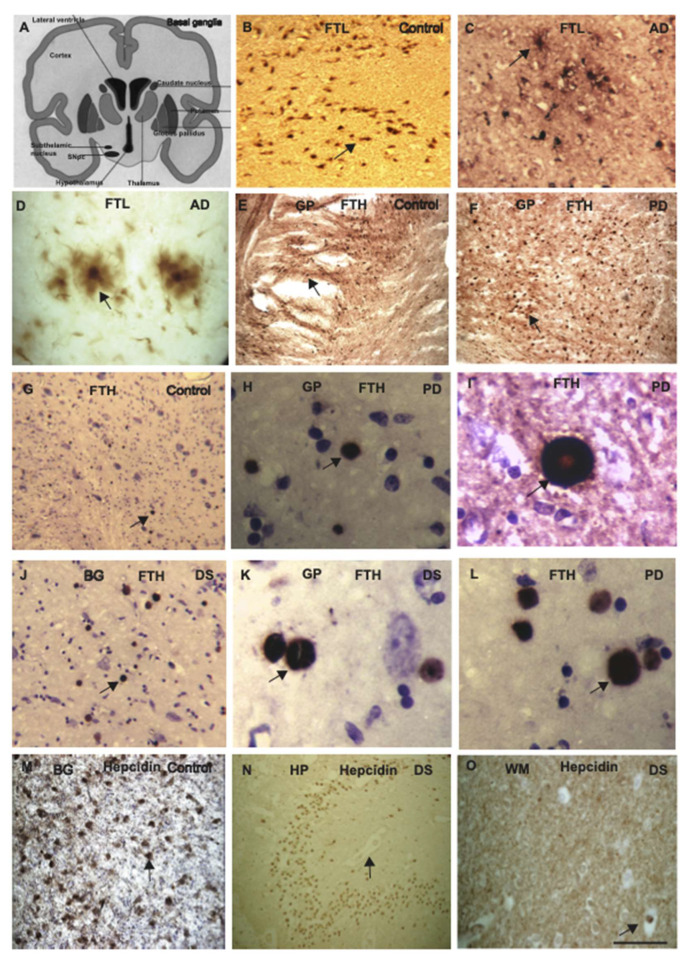
Ferritin light chain and heavy chain proteins localised in the globus pallidus and putamen. To investigate cellular expression of iron proteins, brain sections of AD, PD, and controls were analysed by immunohistochemistry (IHC) using DAB. A cartoon diagram showing the different part of basal ganglia (BG) analysed with DAB staining is presented (**A**). In control brains, in SNpc, a vast amount of granular neuromelanin positive for FTL was found located very close to the blood vessels (**B**); whereas in AD brain, FTL was present in the amyloid plaques (**C**); and on higher magnification it was also present in the microglia (**D**). Brain sections from BG of control and PD subjects when stained with FTH appeared loaded with small vesicles in the thalamus, globus pallidus, and putamen (**E**,**F**), and on higher magnification it appeared like inclusions within the striosome-matrix compartments (**G**–**I**), an appearance very similar to Lewy bodies (**I**–**L**). One control brain section from thalamus was stained with hepcidin and a very similar pattern of iron accumulation was seen, and scattered throughout the thalamus (**M**). In DS brain, hepcidin visible in the blood vessels ((**N**), indicated by arrow), and express in the white matter of internal capsule (**O**). Scale bar: (**B**,**C**,**E**–**G**,**J**,**M**–**O**) = 50 μm; (**D**,**H**,**I**,**K**–**L**) = 10 μm.

**Figure 5 ijms-23-01060-f005:**
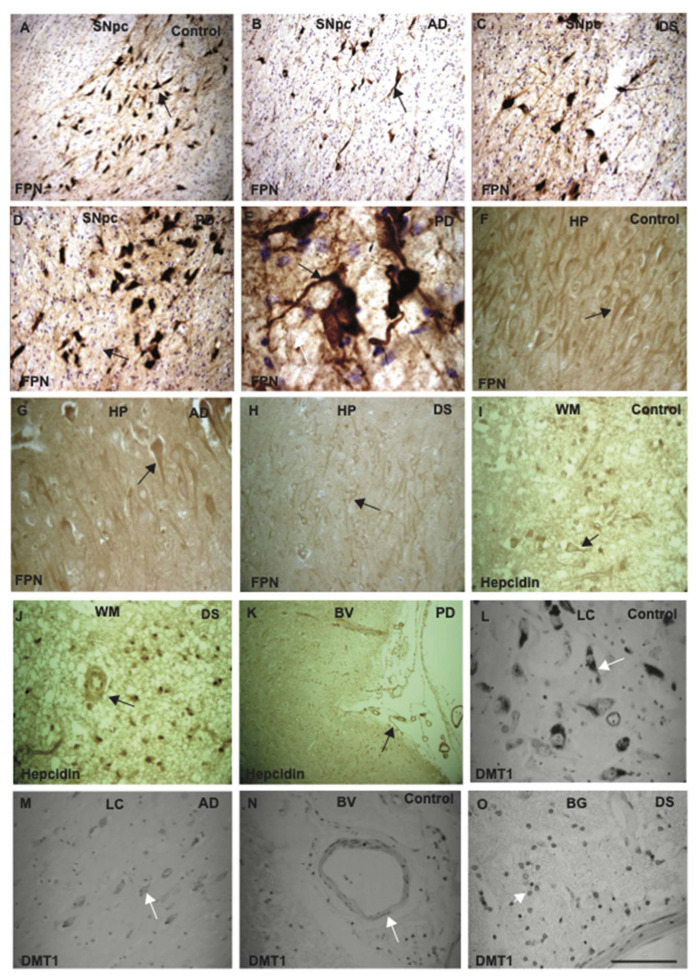
Ferroportin protein expression in the SNpc was higher than in the hippocampus. Ferroportin expression was analysed by IHC using DAB staining. The brain sections from SNpc and BG of PD, AD, DS, and age-matched controls were stained with ferroportin antibody. In controls, AD, and DS brains, the NM cells and long axons and dendrites within SNpc, were stained with ferroportin antibodies (**A**–**C**). In the PD brains, most of the neurons were damaged and ferroportin accumulation was visible in the cell bodies and axon initial segments (AIS) (**D**,**E**). In control brain, FPN expresses in the pyramidal neurons of hippocampus (HP), with strong expression in the cell bodies and in the processes (**F**), but less protein expression seen in the HP of AD and DS brains (**G**,**H**). Hepcidin was visible in the white matter in the oligodendrocytes close to the blood vessels, with stronger expression visible in young DS brains compared to controls (**I**,**J**). In PD brain sections in BG, hepcidin was visible close to the blood vessels (**K**). DMT1 staining was visible in the NM cells of LC with further reduction in AD brain sections (**L**,**M**). DMT1 expressed in the wall of blood vessels and in the endosomes (**N**,**O**). Scale bar: (**A**,**B**,**F**–**K**) = 50 μm, (**C**,**D**,**L**–**O**) = 25 μm, (**E**) = 10 μm.

**Figure 6 ijms-23-01060-f006:**
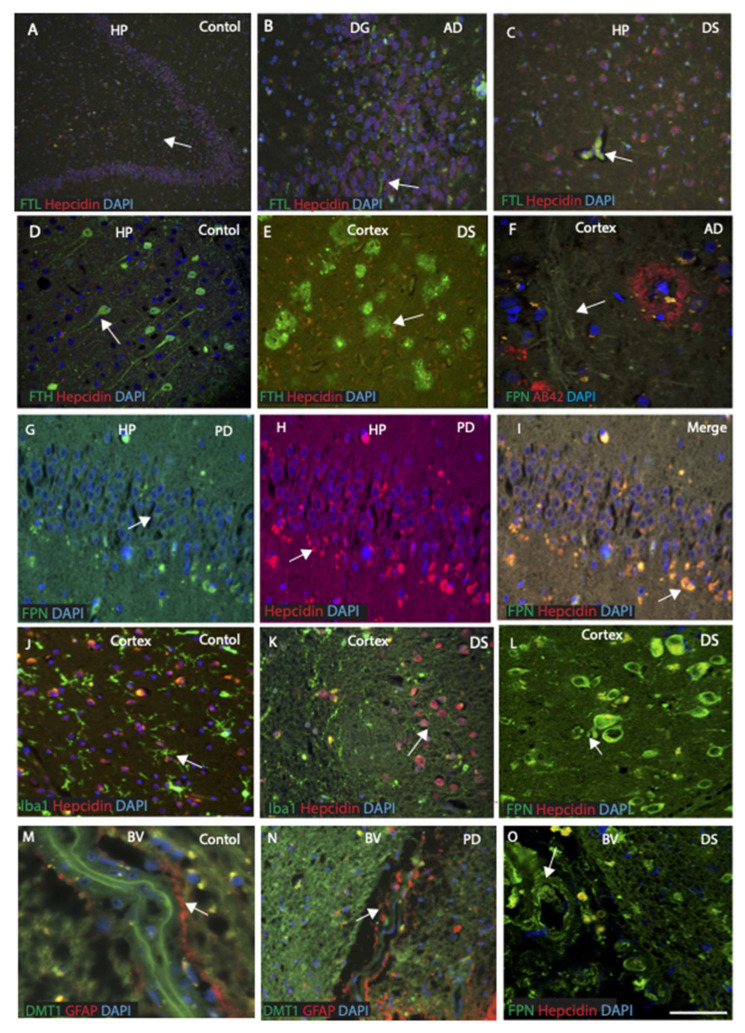
Cellular localisation of ferroportin and hepcidin in the cortex. Fluorescent double staining of hepcidin and FTL was performed using brain sections from cortex and analysed with confocal microscopy. In control brain, hepcidin (red) and FTL (green) were present in the hippocampus and higher magnification in the AD brain showed microglia stained with FTL and hepcidin was located in the granule cell layer (**A**,**B**). Most of the FTL positive microglia (green) were located in the vicinity of the blood vessels and some protein found entering the brain parenchyma from blood vessels (shown with an arrow (**C**). In control brain sections from HP and DG, when stained with FTH, strong FTH staining was visible in the pyramidal neurons (**D**), whereas in DS subjects, FTH was present in the senile plaques and hepcidin in the peripheral cells (**E**). In AD brain, Aβ42, staining was visible in the senile plaques in cortex, but ferroportin was present in the axon/dendrite filaments (**F**). One PD brain (HP) section was stained with ferroportin and hepcidin and analysed with confocal microscope, and both proteins co-localised in the HP (**G**–**I**). Control and DS brain sections were stained with microglia marker (Iba1) and hepcidin. In control brain, ramified microglia (green) and hepcidin (red) were visible without co-localisation (**J**), and in DS brain, activated microglia was present surrounding the plaques while hepcidin was seen in the surviving neurons (**K**). In the DS brain in cortex, ferroportin was expressed in the neurons and there was some localisation with hepcidin (**L**). In control blood vessels, DMT1 expressed in the endothelium of blood vessels and strings of GFAP positive astrocytes appeared probably to receive and transport the proteins into the brain parenchyma (**M**). In PD brain, the blood vessels appeared severely damaged and endothelium significantly losing its structure (**N**). Another DS brain section from BG, when stained with ferroportin and hepcidin, showed both proteins entering in the brain parenchyma from the damaged blood vessel (**O**). Scale bar: (**A**–**C**) = 30 μm; (**D**–**O**) = 20 μm.

**Figure 7 ijms-23-01060-f007:**
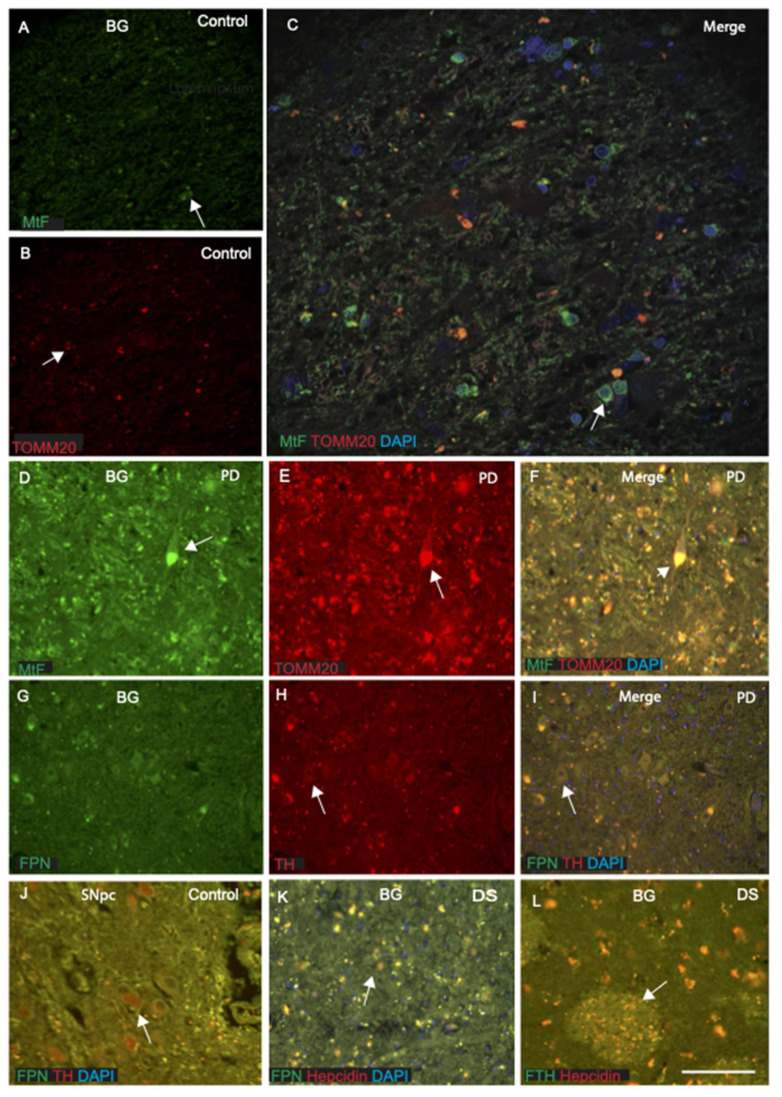
Blood vessel damage and basal ganglia pathology in AD, PD, and ageing brain. Brain sections from basal ganglia were double-labelled with MtF and TOMM20 using IFC and counterstained with DAPI and analysed using confocal microscope. The MtF showed immunoreactivity for the cell bodies and fibres of the caudate nucleus, and some co-localised with TOMM20 (**A**–**C**). On higher magnification, MtF was visible in the neuronal fibres of striatal matrix (**C**). In PD brain section, both proteins co-localised in the striosomes and matrix (**D**–**F**). A control brain section from the SNpc (**J**) and a section from PD brain were stained with ferroportin and tyrosine hydroxylase (TH); in control brain, dopamine cells were filled with TH proteins (red) (**J**), whereas in PD brain, dopamine neurons were empty, but ferroportin staining was visible in the striatal matrix (**G**–**I**). Two basal ganglia sections were stained, ferroportin and hepcidin expressed in the thalamo-striatal fibres in the DS brain (**K**) and fibre bundles were positive for ferroportin. DS brain sections stained with FTH and hepcidin showed patch-like terminal field of the thalamo-striatal projection in the putamen with co-localisation of both proteins (**L**). Scale bar: (**A**–**F**) = 10 μm, (**G**–**L**) = 25 μm.

**Figure 8 ijms-23-01060-f008:**
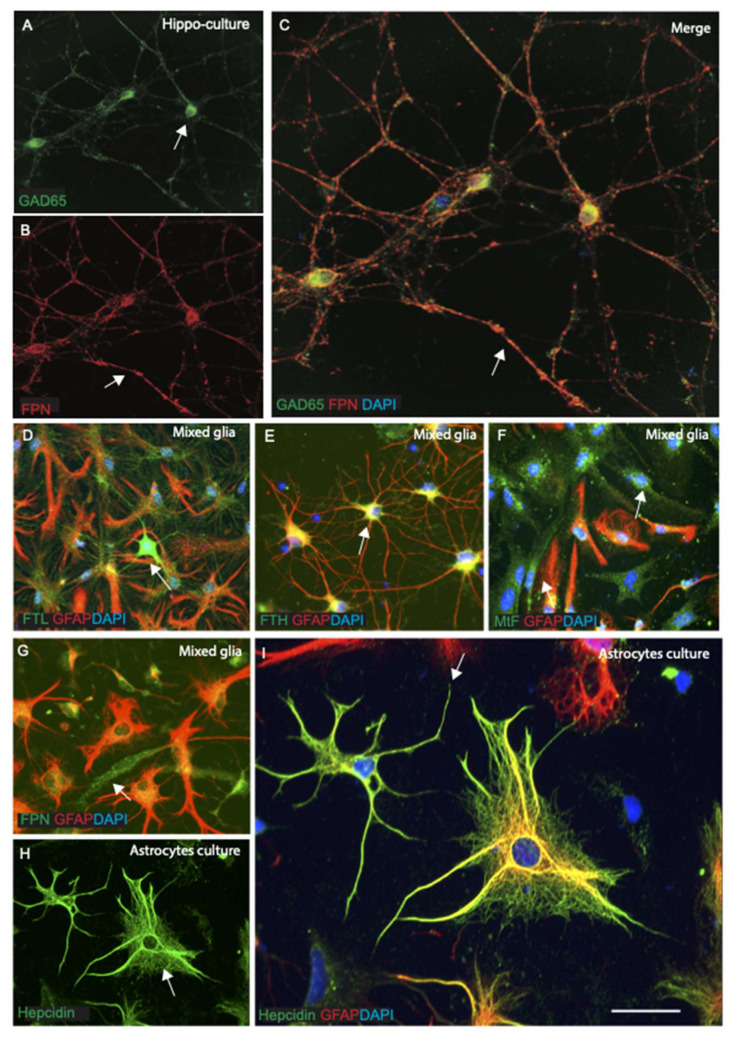
Expression of iron proteins in primary cultures of hippocampal neurons and astrocytes. To investigate the expression of ferroportin in vitro, we grew primary cultures of hippocampal neurons and astrocytes from rat brains. Ferroportin was observed in cultured hippocampal neurons in the nucleus and perinuclear cytoplasm, and in axons/dendrites (in the dendritic spines) and photographed using confocal microscope (**A**–**C**). Hippocampal neurons were co-localised with GAD65 (a marker for GABA) (**C**). Mixed glial cells were stained with each ferritin antibodies and glial fibrillary protein (GFAP) to identify astrocytes (**D**–**F**). FTL was seen in the cell body and in the processes that co-localised with GFAP positive astrocytes (**D**), whereas FTH showed prevalence in the oligodendrocyte-like structures (**E**); a large proportion of mixed glial cells were MtF positive and GFAP negative (**F**). Another slide was stained with ferroportin and GFAP, some bipolar cells were ferroportin positive and GFAP negative (**G**). Astrocytes in culture were stained with hepcidin and GFAP; hepcidin expressed in the cell body and extended into the processes (**H**). Hepcidin expression was present in perinuclear location and as well as in the growth cone (**H**,**I**). Scale bar: (**A**–**C**,**H**,**I**) = 10 μm, (**D**–**G**) = 20 μm.

**Table 1 ijms-23-01060-t001:** Brain sections from Down syndrome, Alzheimer’s disease, Parkinson’s disease, and age-matched controls cases.

Serial No.	Age	Sex F/M	PM Delay	Braak Stage	Cause of Death
**Down syndrome (DS)**					
DS1	56	F	6	6	Not known
DS2	76	F	18	6	Septicaemia
DS3	46	F	8	2	Not known
DS4	52	M	24	6	Bronchopneumonia
DS5	67	M	8	5/6	Alzheimer’s disease (AD), cerebrovascular disease (CVD)
DS6	52	F	-	6	AD, Lewy body dementia
**Alzheimer’s disease (AD)**					
AD1	86	F	86	6	Urinary tract infection/advanced dementia
AD2	88	M	81	6	Urinary tract infection/Addison’s disease and poor immunity/vascular and Alzheimer’s dementia
AD3	83	M	46	6	Bowel ischaemia/hypothyroid/hypertension/Alzheimer’s/atrial fibrillation/chronic kidney disease/vascular dementia
AD4	88	M	22.3	5	Pneumonia/aortic stenosis/mixed dementia/left cerebellar hemisphere haemorrhage
AD5	89	M	44	5	Alzheimer’s disease
AD6	78	M	24	4	Alzheimer’s disease
**Parkinson’s disease (PD)**					
PD1	78	F	37.15	4	Gradual deterioration
PD2	74	M	70.50	3	Bronchopneumonia
PD3	73	F	70.00	5	PD, gradual decline
PD4	73	F	40.15	3	Bronchopneumonia
PD5	66	F	125.30	4	Advanced PD
PD6	71	F	50.10	3	Aspiration pneumonia
**Young Controls (YC)**					
C1	45	F	43.3	0	End stage renal failure/diabetic nephropathy
C2	54	F	10.3	0	Metastatic myxoid liposarcoma/bronchopneumonia
C3	52	F	30.3	1	Bronchogenic cancer
C4	60	F	60	0	Not known
C5	66	M	74	0	Not known
C6	66	F	29.3	2	Metastatic breast cancer
**Old Controls (OC)**					
C7	84	F	81.45	2	Cancer, heart failure
C8	81	M	40.00	1	Chronic obstructive airway disease
C9	85	M	43.35	3	Ca Oesophagus
C10	73	F	28.00	2	Bronchopneumonia, Ca. bronchus
C11	83	F	20.00	1	Bowel resection with complications
C12	88	F	49.25	2	Chronic obstructive airway disease

**Table 2 ijms-23-01060-t002:** Data were analysed by paired Student’s *t*-test (two-tailed) for two group comparison, or by ANOVA test for multiple comparison testing. Values in the figures are expressed as mean ± SEM. A one-way ANOVA was used for comparison of data among control, PD, AD, and DS. Significance was analysed using GraphPad and *p*-values ≤ 0.001 were considered significant, *** = *p* < 0.001, **** = *p* < 0.0001.

Case ID	Numbers	Age (Years)	Gender	Iron Levels in the SNpc (μg/g Wet Weight)	Iron Levels in the LC (μg/g Wet Weight)	*p*-Value
Young control	*n* = 4					Iron levels in LC for all five groups
C1		45	F	179	132	*R*^2^ = 0.49
C2		54	F	192	115	*p* < 0.001
C3		52	F	187	104	*p* = ***
C4		60	F	199	184	
Mean value				=189 μg/g	=133 μg/g	
Older control	*n* = 4					Iron levels in SNpc of AD, PD, and controls
C5		75	F	226	130	*R*^2^ = 0.76
C6		66	F	236	128	*p* < 0.0001
C7		83	M	284	133	*p* = ****
C8		68	M	269	134	
Mean value				254 μg/g	131 μg/g	
Alzheimer’s disease	*n* = 4					Iron levels in SNpc of OC vs. AD
AD1		86	F	280	176	*R*^2^ = 0.69
AD2		88	M	260	192	*p* < 0.001
AD3		83	M	258	188	*p* = ****
AD4		79	F	235	169	
Average				261 μg/g	181 μg/g	
Parkinson’s disease	*n* = 4					Iron levels in SNpc of OC vs. PD
PD1		78	F	280	160	*R*^2^ = 0.84
PD2		74	M	275	158	*p* < 0.0001
PD3		73	F	416	185	*p* = ****
PD4		66	F	367	171	
Mean value				335 μg/g	168 μg/g	
Down’s syndrome	*n* = 4					Iron levels in SNpc of YC vs. DS
DS1		56	F	227	117	*R*^2^ = 0.76
DS3		46	F	225	193	*p* < 0.0001
DS4		52	M	188	164	*p* = ****
DS5		64	M	196	128	
Average				209 μg/g	150 μg/g	

## Data Availability

All data showed in the Table 1 and Table 2.

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
