# Peer review of "Interplay of Ferritin Accumulation and Ferroportin Loss in Ageing Brain: Implication for Protein Aggregation in Down Syndrome Dementia, Alzheimer’s, and Parkinson’s Diseases"

_ijms, 2022, doi:10.3390/ijms23031060_

Round 1

Reviewer 1 Report

Comments are attached

Author Response

Manuscript ID: ijms-1515626

Type Article

Manuscript title: Interplay of ferritin accumulation and ferroportin loss in ageing brain: implication for protein aggregation in Down syndrome dementia, Alzheimer’s and Parkinson’s diseases.

Recommendation: Major revision invited

Reviewer #1: Comments and Suggestions for Authors

  1. In the title of the article, the term “Downs syndrome” should be replaced by “Down syndrome”. I am surprised that the term characterizing the disease itself (as part of a medical investigation) is used incorrectly throughout the text of the manuscript. The correct term should be used.

Response 1: Thank you for your suggestion and in this revised manuscript, we changed to the term “Down syndrome” throughout the manuscript. There is an international agreement of the term to use “Down syndrome” or “Down’s syndrome” made very confusing as paper was edited by all co-authors. Now we finally agreed to use the term “Down syndrome” in this manuscript and will follow in near future. In current revised manuscript, any changes added, as you adviced are highlighted in our responses bellow.

  1. In the Abstract, the statement “Ferritin, an iron storage protein, deposited in senile plaques in AD 25 and DS brain, as well as in neuromelanin-containing neurons in Lewy bodies in PD brain.” is not understood. Maybe, there is a verb missing there (!), thereby necessitating restructuring of the phrase. At any rate, the statement needs to be rephrased.

Response 2: Changed to “Ferritin, is an iron storage protein, that deposited in senile plaques in AD and DS brain, as well as in neuromelanin-containing neurons in Lewy bodies in PD brain.”

  1. In the introduction section, the statement “Another neurological disease is Downs syndrome (DS) dementia, also develop the clinical features of AD due to the presence of an extra copy of chromosome 21 (Hsa21) where the APP gene is located (29-32).” is long and at the end incomprehensible. It should be split in two sentences and grammatically adjusted so as to project meaning.

Response 3: We split it in two sentences and thanking you for your suggestion. “Another neurological disease is Down syndrome (DS) dementia, also develop the clinical features of AD. This is due to the presence of an extra copy of chromosome 21 (Hsa21) where the APP gene is located (29-32).

  1. In the same section, the statement “Aβ plaque deposition is an early event seen in DS brain, Aβ and the iron storage protein ferritin have been shown to co-localize in the vascular amyloid deposits of plaque in post-mortem AD/DS brains (26, 38-40).” should be corrected to read “Aβ plaque deposition is an early event seen in DS brain. Aβ and the iron storage protein ferritin have been shown to co-localize in the vascular amyloid deposits of plaques in post-mortem AD/DS brains (26, 38-40).”.

Response 4: We changed it to as you advised. “Aβ plaque deposition is an early event seen in DS brain. Aβ and the iron storage protein ferritin have been shown to co-localize in the vascular amyloid deposits of plaque in post-mortem AD/DS brains (26, 38-40).”

  1. A couple of lines below, the statement “The DA neurons of SNpc extends their fibres to the caudate-putamen forming the nigrostriatal pathway and this pathway is essential for voluntary movements in normal subjects (43, 44).” should be rewritten to read “The DA neurons of SNpc extend their fibres to the caudate-putamen, forming the nigrostriatal pathway. This pathway is essential for voluntary movements in normal subjects (43, 44).”.

Response 5: We changed it to as you advised. “The DA neurons of SNpc extends their fibres to the caudate-putamen, forming the nigrostriatal pathway. This pathway is essential for voluntary movements in normal subjects (43, 44).

  1. In a number of figures the contents of the legends provided is essentially the same as the articulation of the work done in the text. The legends should be modified so as to succinctly present the contents of the work depicted in the pictures provided.

Response 6: We have shortened and modifies the figure legends as you advised.

  1. In section 2.8, the statement “To evaluate any damaged in the PD hippocampus, one PD brain (HP) section was stained with ferroportin and hepcidin and analysed with confocal microscopy.” should be corrected to read “To evaluate any damage in the PD hippocampus, a PD brain (HP) section was stained with ferroportin and hepcidin and analysed through confocal microscopy.”.

Response 7: Changed to as you recommended. “To evaluate any damage in the PD hippocampus, a PD brain (HP) section was stained with ferroportin and hepcidin and analysed through confocal microscopy.”

  1. In the discussion section, the statement “However, with ageing, the brain becomes more susceptible and numerous factors (hypertension, abnormal glucose metabolism and oxidative stress) could alter endothelium elasticity, leading to a leaky BBB (57-59).” pertains to the susceptibility of the aging brain. What does the brain become susceptible to?

Response 8: The sentence changes to,

“However, with ageing, the brain becomes more susceptible and numerous factors (hypertension, abnormal glucose metabolism and oxidative stress) could alter endothelium elasticity, leading to a leaky BBB. The leaky BBB, allows to enter many macromolecules and cytokine/chemokines in the brain parenchyma that lead to neuroinflammation (57-59). “

  1. In the ensuing paragraphs, the statement “We have shown by biochemical analysis of non-haem iron level in SNpc and LC in AD, PD, DS and age matched normal subjects.” is not complete and therefore not understood. Essentially, there is no sentence. It should be rewritten correctly so that it conveys a specific message.

Response 9: The sentence changes to, “We have shown by biochemical analysis of non-haem iron level in SNpc and LC was much higher in PD compared to AD and older controls (Figure 1A-B). In young control subjects (YC) and in young DS subjects, iron level was significantly lower compared to older subjects (Figure 1A). These findings suggest that iron accumulation in neuromelanin, particularly in the SNpc is age-related and even could be disease related like in PD.”

  1. In the follow up paragraph, the statement “It was also reported that ferroportin is involved in transporting calcium and other metals (65) and in this study, we are report for the first time that AIS may be implicated in metal ion exchange in some selective neurons.” should be rewritten to read “It was also reported that ferroportin is involved in the transport of calcium and other metals (65). In this study, we report for the first time that AIS may be implicated in metal ion exchange in certain selective neurons.”. The statement is too long and the reader is lost.

Response 10: We change the sentence as you advised:

“It was also reported that ferroportin is involved in transporting calcium and other metals (65).  In this study, we report for the first time that AIS may be implicated in metal ion exchange in certain selective neurons.”

  1. Further into the discussion section, the statement “We would like to suggest that the ferroxidase property of FTH could be protect the different types of projection neurons by providing adequate nutrients.” should be amended so that it is comprehensible. A proposal would be the following: “We would like to suggest that the ferroxidase property of FTH could be to protect the different types of projection neurons by providing adequate nutrients.”.

Response 11: We change the sentence as you advised:

“We would like to suggest that the ferroxidase property of FTH could be to protect the different types of projection neurons by providing adequate nutrients.”

  1. Further down, the statement “It is also an important retrograde solute carrier, not only binding with iron, but also with other metals (calcium, manganese, cupper, cadmium and aluminium) (78, 79).” should corrected to read “It is also an important retrograde solute carrier, not only binding iron, but also other metals (calcium, manganese, copper, cadmium and aluminium) (78, 79).”.

Response 12: Thanking you for your suggestion. “It is also an important retrograde solute carrier, not only binding iron, but also with other metals (calcium, manganese, cupper, cadmium and aluminium) (78, 79).”

  1. In the last paragraph of the discussion, the statement “Contrastingly, in astrocytes culture, hepcidin not only expressed in the cell body but also extended in to the processes and as well as in the growth cones.” should be corrected to read “In contrast to that, in astrocyte cultures, hepcidin is not only expressed in the cell body but expression also extends into the processes and growth cones as well.”.

Response 13: We change the sentence to “In contrast to that, in astrocytes culture, hepcidin not only expressed in the cell body but expression also extends in to the processes and growth cones as well as.”

  1. In the conclusion section, the statement “This imbalance in brain iron homeostasis may leads oxidative stress in cells, finally causing cell death.” should be grammatically corrected to read “This imbalance in brain iron homeostasis may lead to oxidative stress in cells, finally causing cell death.”.

Response 14: The sentence changed to, “This imbalance in brain iron homeostasis may lead to oxidative stress in cells, finally causing cell death.”

  1. In section 4.5 (Materials and methods), ferric iron is FeCl3. By the same token, hydrochloric acid is HCl.

Response 15: In section 4.5 (Materials and methods), ferric iron is change to FeCl3

We would like to thanks both reviewers for their suggestions which are helping to improve this manuscript.

Reviewer 2 Report

  • Table 1: Sex and Age parameters are swapped.
  • Figure 3. Western blots are poor and, as it stands, unacceptable for publication. Bands are clearly overexposed e.g., Actin, Hepcidine, DMT1, FTH. Actin bands are unequal, clearly indicating uneven loading of total proteins. Protein levels should be compared in relation to Actin, and in the particular case of blood proteins such as Ferritin it is important to also stain and compare against Albumin. 
  • The authors fail to relate their findings with the recent understanding of PD/AD pathophysiology. It has been elegantly shown that amyloid “seeds” e.g., asyn and tau fibrils, can trans-synaptically spread (PMID: 31254094; PMID: 33632316; PMID: 31249873) between connected neuronal populations, inclusively from the PNS to the brain, while provoking PNS impairment and degeneration. This might elucidate for instance why a PD patients often display GI and nociceptive impairment long before the “classic” motor symptomatology and CNS dopaminergic loss.
  • Further, the existence of diverse amyloid conformations or “strains” is most likely responsible for the clinical heterogeneity found in PD & AD (PMID: 33978813; PMID: 26324905; PMID: 30497516). The authors would greatly enhance the overall relevance of their conclusion if they acknowledge these breakthrough findings, and speculate on the role of ferritin accumulation and ferroportin loss in the prion-like propagation of amyloids e.g. asyn and tau aggregates, in CNS aging and onset of PD and AD.

Author Response

Manuscript ID: ijms-1515626

Type Article

Manuscript title: Interplay of ferritin accumulation and ferroportin loss in ageing brain: implication for protein aggregation in Down syndrome dementia, Alzheimer’s and Parkinson’s diseases.

Recommendation: Major revision invited

Reviewer #2:  Comments and Suggestions for Authors

Thank you for your comment and suggestion and we have tried to edited the manuscript. 

  • Table 1: Sex and Age parameters are swapped.

Response 1: Thank you for your suggestion and in this revised manuscript we edited the table and change the sex and age in right order.

  • Figure 3. Western blots are poor and, as it stands, unacceptable for publication. Bands are clearly overexposed e.g., Actin, Hepcidine, DMT1, FTH. Actin bands are unequal, clearly indicating uneven loading of total proteins. Protein levels should be compared in relation to Actin, and in the particular case of blood proteins such as Ferritin it is important to also stain and compare against Albumin. 

Response 2: We have changed some panels of Western blot data. In these WB experiments, Protein lysates were prepared from basal ganglia and SNpc of human brains from PD, DS and age matched controls. First of all, human brain samples were very heterogeneous, due to age, disease and other factors as described in Table 1. In human brain (basal ganglia and SNpc) contain many different types of cells in the grey matter, in white matter and in the blood vessels. These samples are very different from serum or homogeneous tissue samples (e.g. liver or muscles) or cell line lysate used for analysis. FTL, FTH and MtF blots repeated in all PD, DS and age matched controls (N = 6 in each group) and results were consistence. MtF levels were much lower then FTL and FTH. MtF blot was re-probed with Albumin as suggested by Reviewer #2. FPN and DMT1expresses in the blood vessels wall and difficult to obtain equal amounts of membrane bound proteins from different brain samples. We have a great deal of experience to obtain right amount of membrane bound protein from total brain lysate. To verify the expression pattern of these two proteins (ferroportin and DMT1) was further analysed with immunohistochemistry. Hepcidine, transferrin and Ubiquitin blots were loaded same amounts samples, in three separate gels and probed by three different antibodies same day, by same scientist. There was no heterogeneity in loading in these gels and results are consistent. Transferrin gel was re-probed with b-actin. We replaced the previous b-actin blot panel. I hope you will agree and consider this blot and figure 3A is acceptable for publication.

  • The authors fail to relate their findings with the recent understanding of PD/AD pathophysiology. It has been elegantly shown that amyloid “seeds” e.g., asyn and tau fibrils, can trans-synaptically spread (PMID: 31254094; PMID: 33632316; PMID: 31249873) between connected neuronal populations, inclusively from the PNS to the brain, while provoking PNS impairment and degeneration. This might elucidate for instance why a PD patients often display GI and nociceptive impairment long before the “classic” motor symptomatology and CNS dopaminergic loss.

Response 3: In this manuscript, we mainly focused on human subjects, comparing
the distribution and expression of key iron proteins ferritin, ferroportin, DMT1 and hepcidin in human brain tissues from patients with AD, DS, PD and age-matched controls. This manuscript mainly described neuropathological findings in AD, DS and PD particularly in basal ganglia and do not describe tauopathy, multiple system atrophy or neuropathic pain as suggested in (PMID: 33632316 and PMID: 33978813).

We are aware of prion like propagation of a-synuclein and tau fibril could trans-synaptically from peripheral nervous system to central nervous system.  We read all three papers (PMID: 31254094; PMID: 33632316; PMID: 31249873) and add a paragraph in the discussion section discussing the relevant points that fits with this manuscript (from PMID: 31254094; PMID: 26324905) coted as references 83 and 84. The sentence added in the discussion as: “Several groups recently reported trans-synaptic propagation of a-synuclein protein in mouse and rat models (83, 84). In rat model, authors proposed bidirectional a-synuclein propagation via the vagus nerve, i.e., duodenum-to-brainstem-to-stomach (83). However, we haven’t seen any synuclein like protein propagation from either heart or muscle to the brain parenchyma in control or disease brains. The animal models are very useful to investigate functional characteristics but cannot recapitulate all pathological and clinical features observed in human neurodegenerative diseases.”

  • Further, the existence of diverse amyloid conformations or “strains” is most likely responsible for the clinical heterogeneity found in PD & AD (PMID: 33978813; PMID: 26324905; PMID: 30497516). The authors would greatly enhance the overall relevance of their conclusion if they acknowledge these breakthrough findings, and speculate on the role of ferritin accumulation and ferroportin loss in the prion-like propagation of amyloids e.g. asyn and tau aggregates, in CNS aging and onset of PD and AD.

Response 4: In this manuscript, we described AD, PD and DS, brain iron homeostasis shows very different patterns in different brain compartments.  In conclusion, our work indicates that PD is a complicated and multifactorial disorder with varying brain pathology when compared to AD or DS dementia. We included another line in the discussion as you suggested on tau protein propagation described by Wegmann et al, 2019 (PMID: 31249873) as below.

“Similarly, another paper investigated the tau spread by expressing human tau via viral vector
in old and young mice. The old mouse showed increased tau spreading in the hippocampus and concluded that age-related brain region specific tau spreading could be a related risk for
sporadic AD (ref 85,
PMID: 31249873). We do support the hypothesis of age-related risk for sporadic AD, as we have noticed a strong link between age-associated increase in iron stores in the brain alongside failure of clearing processes, and an increasing incidence of AD in DS subjects with advancing age.”

We would like to thanks both reviewers for their suggestions which are helping to improve this manuscript.

Round 2

Reviewer 2 Report

The authors mostly failed to addressed my comments and suggested references. The purpose to is to "fit" them to the author's findings, but exactly the opposite: how does the the interplay of ferritin accumulation and ferroportin loss in ageing brain impact on the current view of proteinopathies? Also the author's fail to acknowledge the importance of preclinical research using widely recognised rodent models of disease that can be used to study shared mechanisms to an extent that is simply not possible to do in patients.

For these reasons, I can not, at this point recommend the current version for publication.